# Associations between common genetic variants and income provide insights about the socio-economic health gradient

We conducted a genome-wide association study on income among individuals of European descent ($N$ = 668,288) to investigate the relationship between socio-economic status and health disparities. We identified 162 genomic loci associated with a common genetic factor underlying various income measures, all with small effect sizes (the Income Factor). Our polygenic index captures 1–5% of income variance, with only one fourth due to direct genetic effects. A phenome-wide association study using this index showed reduced risks for diseases including hypertension, obesity, type 2 diabetes, depression, asthma and back pain. The Income Factor had a substantial genetic correlation (0.92, s.e. = 0.006) with educational attainment. Accounting for the genetic overlap of educational attainment with income revealed that the remaining genetic signal was linked to better mental health but reduced physical health and increased risky behaviours such as drinking and smoking. These findings highlight the complex genetic influences on income and health.

Income is a crucial determinant of individuals' access to resources and overall quality of life. Extensive evidence shows that income is positively correlated with subjective well-being, overall health and life expectancy[1-5]. For instance, the gap in life expectancy between the richest and poorest 1% of individuals in the USA has been estimated to be 14.6 years for men (95% confidence interval (CI), 14.4 to 14.8 years) and 10.1 years for women (95% CI, 9.9 to 10.3 years)[6]. Notably, higher income is associated with increased longevity and well-being across the entire income distribution, highlighting its broad relevance in current society[3,6,7].

Income is a complex phenotype influenced by many factors, including environmental conditions and education[8,9]. Parents' socio-economic status (SES) shapes a child's developmental trajectory, including their skills, behaviours, educational attainment (EA), career prospects and eventual adult income[10,11]. Moreover, certain heritable individual characteristics, such as cognitive ability and personality traits[12-14], are well-known predictors of income within contemporary societies in Europe, North America and Australia. Twin studies have estimated income heritability in these societies to be around 40–50% (refs. 15–17). However, the heritabilities of income and its associated

genes are not fixed; rather, they reflect social realities shaped by technological, institutional and cultural factors[18]. These factors are malleable and vary across different regions and historical epochs, which can lead to fluctuations in heritability estimates for SES over time[19,20] and imperfect genetic correlations across samples[21].

The results from statistically well-powered genome-wide association studies (GWASs) of SES present numerous opportunities to shed light on these social realities. For example, they allow investigating questions about sex differences in labour market processes, cross-country comparisons of the genetic architecture of income and investigating the processes contributing to intergenerational social mobility[22]. They also facilitate studies investigating the interaction effects between genetic and environmental factors. Furthermore, they enable the exploration of genetic correlations between income and health outcomes, potentially unveiling new insights into the positive relationship between socio-economic status and health outcomes (the socio-economic health gradient).

Two previous GWASs have been conducted on household income[23,24]. The first was in a sample of 96,900 participants from the initial release of the UK Biobank (UKB)[25] and found two loci. The second

✉e-mail: a.abdellaoui@amsterdamumc.nl; David.Hill@ed.ac.uk; p.d.koellinger@vu.nl

**Table 1 | GWAS summary**

| Measure | N | Proportion female | No. of SNPs | Mean $\chi^2$ | No. of loci | $h^2$ (s.e.) |
|---|---|---|---|---|---|---|
| Household | 497,413 | 0.55 | 11,500,222 | 1.54 | 41 | 0.06 (0.003) |
| Individual | 72,601 | 0.54 | 5,986,804 | 1.06 | 0 | 0.04 (0.007) |
| Occupational | 443,064 | 0.57 | 11,500,419 | 1.64 | 59 | 0.08 (0.003) |
| Parental | 128,724 | 0.50 | 6,144,179 | 1.11 | 1 | 0.05 (0.006) |
| Income Factor | 668,288[a] | – | 9,131,507 | 1.94 | 162 | 0.07 (0.002) |

The Income Factor is derived from a meta-analysis across the four income measures: individual, occupational, household and parental. The mean $\chi^2$ was computed only with the HapMap 3 SNPs. The number of approximately independent loci (sixth column) was obtained using FUMA. The SNP heritability ($h^2$) was estimated with LDSC. [a]The estimated effective sample size is reported for the Income Factor. Some individuals contributed multiple times to different income measures.

was carried out in the full release of the UKB with 286,301 individuals and found 30 approximately uncorrelated loci. A meta-analysis of these results with the genetically correlated trait EA increased the effective sample size to 505,541 individuals and identified 144 loci. A recent GWAS on occupational status in the UKB data identified cognitive skills, scholastic motivation, occupational aspiration, personality traits and behavioural disinhibition (proxied by attention deficit hyperactivity disorder) as potential mediating factors linking genetics to occupational status[26].

Building on these earlier contributions, we conducted a GWAS leveraging multiple income measures. We ran sex-stratified analyses and meta-analysed results from 32 cohorts across 12 countries (Australia, Croatia, Denmark, Estonia, Finland, Germany, the Netherlands, Norway, Sweden, Switzerland, the United Kingdom and the USA) and three continents, yielding the largest GWAS on income to date with an effective sample size of 668,288 (Table 1). Due to data availability and statistical power considerations, our analyses and conclusions are restricted to individuals carrying genotypes most similar to the EUR panel of the 1000 Genomes dataset, as compared with individuals sampled elsewhere in the world (1KG-EUR-like individuals).

The greater statistical power of our GWAS enabled us to conduct a series of follow-up analyses that investigate the socio-economic health gradient from a genetic perspective. In particular, we leveraged the data to compare the GWAS results for income and EA to disentangle their unique genetic correlates with health. Furthermore, our multi-sample approach and sex-specific GWAS results allowed us to test for possible differences in the genetic architecture of income across samples and sexes.

For a less technical description of the paper and how it should—and should not—be interpreted, see the Frequently Asked Questions (FAQ) section in the Supplementary Information and Box 1.

## Multivariate GWAS of income
### GWAS of four different measures of income
We used four measures of income (individual, occupational, household and parental income) and conducted a GWAS meta-analysis of their shared genetic basis (Table 1). Supplementary Information Section 2.1 discusses the differences between these measures and their relative advantages and disadvantages as proxies for individual income. Dropping parental income from the meta-analysis leads to a slight statistical power decrease but does not qualitatively change our results.

A sex-stratified GWAS was carried out on each available income measure in each cohort, using at least the first 15 genomic principal components (PCs) to control for population stratification. Inflation, business cycle, age effects and other potential confounds were controlled for at the cohort level by using dummy variables (see the preregistered analysis plan, section 6, at https://osf.io/rg8sh/). We restricted our analyses to 1KG-EUR-like individuals who were not currently enrolled in an educational programme or who were aged above 30 if their current enrolment status was unknown. The natural

log transformation was applied to the income measures. We applied standardized quality control procedures to each cohort-level result (see Supplementary Information Section 2.4 for details). For each sex and each income measure, we performed a sample-size-weighted meta-analysis with METAL[27]. We then meta-analysed the male and female results for each income measure using the multi-trait analysis of genome-wide association summary statistics method (MTAG)[28], which accounts for any potential genetic relatedness between the male and female samples.

The four income measures' pairwise genetic correlation ($r_g$) estimates demonstrated substantial shared genetic variance, with all pairwise $r_g$ values at least 0.8 (Fig. 1).

### The Income Factor
Next, we meta-analysed the association results across the four income measures using MTAG (see Supplementary Information Section 2.5 for details). We observed that the MTAG procedure yields nearly identical results to the common factor function in genomic structural equation modelling[29]. Thus, we hereafter refer to the meta-analysed income as 'the Income Factor'. Since MTAG already applies a bias correction with the intercept from linkage disequilibrium score regression (LDSC)[30], we did not apply further adjustments for cryptic relatedness and population stratification.

The Income Factor GWAS was estimated to have an effective sample size ($N_{eff}$) of 668,288, on the basis of occupational income's heritability scale ($N_{eff} = 1,198,347$ on the basis of individual income). The genetic correlation between individual income and the Income Factor is indistinguishable from 1 (Fig. 1).

### Identification of genetic loci
Across the four GWASs on different income measures, we identified 86 non-overlapping loci in the genome (see Supplementary Information Section 2.6 for the definition of loci and lead single nucleotide polymorphisms (SNPs), and Extended Data Fig. 1a for the distribution of associated loci across the four income traits). Table 1 summarizes the results. Occupational and household income showed the most genetic associations (59 and 41 loci, respectively), as expected on the basis of sample sizes and SNP-based heritability estimates based on LDSC (occupational: $h^2 = 0.08$, s.e. = 0.003; household: $h^2 = 0.06$, s.e. = 0.003). Gene-based analysis was performed on the genes that overlapped with each locus using multi-marker analysis of genomic annotation (MAGMA), where 102 attained genome-wide significance, with 63 being unique to occupational income, 24 unique to household income and 55 shared between the two. No other genes attained statistical significance (Extended Data Fig. 1b).

The meta-analysis across the income measures led to a substantial increase in power, which allowed us to identify 162 loci tagged by 207 lead SNPs (Fig. 2). Of these loci, 88 were newly identified compared with the previously published GWAS household income result conducted in the UKB[24]. The genetic correlation of the previous household income GWAS result was 0.92 (s.e. = 0.008) with the Income Factor

## BOX 1

# Understanding genetics and income: a cautionary overview

Given the frequent misunderstanding of research on genetics and human behaviour, it is important to recognize the complexities underlying connections between genes and social outcomes and to communicate what our findings mean clearly and with appropriate nuance.

### What did we do and why?

Several types of 'luck' help shape an individual's life trajectory, such as their society of birth, their parents and the genetic variants they inherit. Our study captures elements of this by examining the relationship between millions of genetic variants and income through a GWAS. GWASs of income can provide valuable insights into the genetic factors associated with income and how they interact with environmental factors, enhancing our understanding of intergenerational mobility and socio-economic disparities.

GWASs of income can shed light on societal processes that favour certain genetic predispositions, providing insights into our socio-economic system as well as into the relationships between income and health disparities. Recent GWASs have shown that socio-economic outcomes share genetic overlap with various health outcomes, with a considerable portion mediated through social environments[57].

### What did we find?

We identified numerous genetic variants associated with income, each with minor effects but collectively correlating with education, cognition, behaviour and health. We found notable differences between income and EA in their genetic associations with health outcomes. For several psychiatric disorders—namely, autism, schizophrenia and obsessive-compulsive disorder—the genetic relationships acted in opposing directions. Shared genetic effects between income and health may stem from various causes. Genes might affect both income and health. Alternatively, higher income could lead to better health outcomes, not only directly but also indirectly through improved living conditions from family members or neighbourhoods. Conversely, existing health problems may limit income opportunities, potentially due to reduced work capacity or increased health-care costs.

When predicting differences between siblings, the overall predictive strength of these genetic effects diminishes substantially—by approximately 75%. Possible explanations for this are that the direct causal effects of the genetic variants are smaller than the causal effects of environmental factors that correlate with these genetic variants (for example, the effects of parental nurture on their children) and that the way parents resemble each other (assortative mating) magnifies the predictive power of genetic effects.

We observed some variability in the genetic factors influencing income across the Western countries we analysed and between genders, underscoring that the genetic associations we report here should not be interpreted as fixed or universal.

### Neither genetic nor environmental determinism is warranted

Historically, misinterpreting the role of genetics in shaping social outcomes has occasionally fuelled controversial ideologies with far-reaching consequences. It is important to mitigate the risk of such misunderstandings, particularly the notions of genetic or environmental determinism. In this context, we emphasize the following:

One's genetic makeup or the family and societal environment into which one is born does not dictate one's intrinsic value. The genetic variants that matter for income, and their effects, depend on the environment—that is, on what skills are valued by the labour market and by society. As the labour market changes or as government policies change, so can the variants and their effects.

It is important to recognize how genetics can impact income through diverse pathways, affecting one's own or one's parents' health, cognition, skills and productivity-related behavioural tendencies, such as creativity, risk taking and adaptability. Additionally, genetics can influence characteristics favoured or discriminated against in the labour market due to societal preferences.

As with previous genetic studies on social outcomes such as EA, the findings of this study have limited generalizability across different populations.

and 0.94 (s.e. = 0.006) when we restricted our analysis to only our household income measure.

Furthermore, we conducted conditional and joint association analysis using the 207 lead SNPs associated with the Income Factor[31], revealing 57 secondary lead SNPs ($P \le 5 \times 10^{-8}$). Of these secondary lead SNPs, 55 were located within the original primary genomic loci (Supplementary Table 30 and Supplementary Information Section 2.6).

### Effect sizes

The effect sizes of the lead SNPs were small across all analyses. For example, when we adjusted for the statistical winner's curse in the Income Factor results, one additional count in the effect allele of the median lead SNP was associated with an increase in income of 0.30%. These effect size calculations require an assumption about the standard deviation of the dependent variable because MTAG yields standardized effect size estimates; we used the standard deviation estimate of log hourly occupational wage from the UKB, which is 0.35. The estimated effects at the 5th and 95th percentiles of the SNP effect size distribution were 0.18% and 0.60%, respectively (Supplementary Information Section 2.7). To put these estimates into perspective, the median

annual earnings of full-time workers in the USA was U$56,473 in 2021[32]. A 0.3% increase would equal an additional annual income of US$169 (95% CI, US$102 to US$339). In terms of the variance explained ($R^2$), all of the lead SNPs had $R^2$ lower than 0.011% after adjustment for the statistical winner's curse (Supplementary Fig. 1).

### Cross-sex and cross-country heterogeneity

The heritability of income and its genetic associations may vary across different social environments or different groups within an environment. To investigate the potential heterogeneity of genetic associations with income, we examined cross-cohort genetic correlations. We found that the inverse-variance-weighted mean genetic correlations across pairs of cohorts were 0.45 (s.e. = 0.22) for individual income, 0.52 (s.e. = 0.13) for household income and 0.90 (s.e. = 0.24) for occupational income (Supplementary Table 28a–c).

Next, we meta-analysed cohorts from the same country with the same income measure available and estimated the genetic correlations across these countries (Estonia, the Netherlands, Norway, the United Kingdom and the USA; Extended Data Fig. 2a). For most country pairs, the genetic correlation of the same income measure

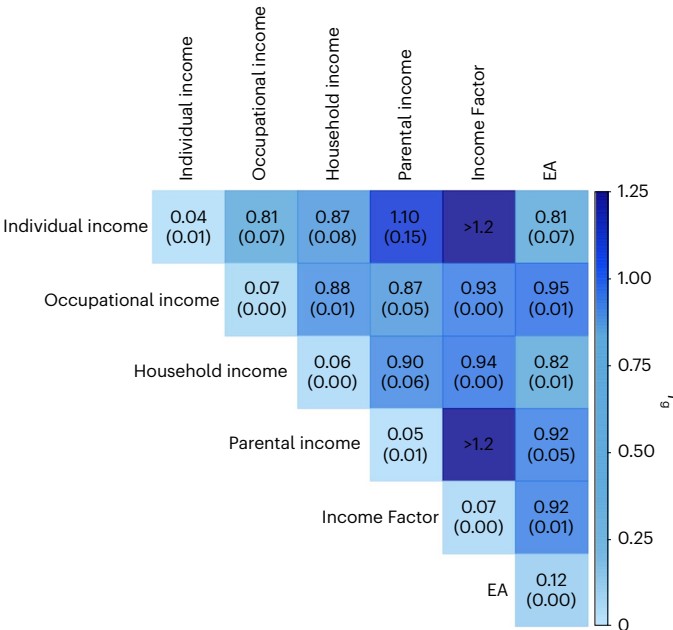

**Fig. 1 | Genetic correlations between income measures.** LDSC estimates of pairwise genetic correlations between the four input income measures, the meta-analysed income (Income Factor) and EA. The diagonal elements report SNP heritabilities from LDSC. The standard errors are reported in parentheses. Some of the results were out-of-bound estimates (exceeding 1.2).

is >0.8. While meta-analysis increases statistical power and yields more precise estimates of the average effect size, it also tends to mask non-random heterogeneity in effect size estimates across samples. Despite this latter point, we found that occupational income in Norway displayed lower genetic correlations with occupational or household income in other countries, ranging from 0.43 (s.e. = 0.23) to 0.82 (s.e. = 0.10). Similarly, occupational income's genetic correlation with EA was also lower in Norway ($r_g$ = 0.69, s.e. = 0.08) than in the other countries. These findings align with phenotypic evidence that ranks Norway the lowest among Organisation for Economic Co-operation and Development countries in terms of financial returns for obtaining a college degree[33].

We then investigated whether the large number of samples from the United Kingdom in our meta-analysis could have skewed our results. To address this, we conducted a separate meta-analysis procedure for the UK and non-UK cohorts, comprising participants from ten countries. We obtained two distinct GWAS results for the Income Factor and found a perfect genetic correlation of 1.001 (s.e. = 0.03) between them. Thus, the average effect sizes of SNPs associated with the Income Factor are almost identical in UK and non-UK cohorts.

We observed slight between-sex heterogeneity in the genetic associations of income, as supported by the evidence presented in Extended Data Fig. 2b. The estimated between-sex genetic correlations based on meta-analysed GWAS results for individual, occupational and household income were 1.06 (s.e. = 0.32), 0.91 (s.e. = 0.03) and 0.95 (s.e. = 0.03), respectively. Notably, the latter two estimates were statistically distinguishable from unity but remained above 0.9. Most cohort-specific cross-sex genetic correlations for income are too noisy to be interpreted (Supplementary Table 17b–d). One exception is the UKB sample, which shows a non-perfect genetic correlation between men and women for occupational income ($r_g$ = 0.91, s.e. = 0.03). Another exception is the Danish iPsych cohort, where we estimated a genetic correlation of 0.76 (s.e. = 0.10) between maternal and paternal income. These findings are consistent with the hypothesis that men and women face non-identical labour market conditions. The lower genetic correlation between maternal and paternal income suggests

that differences in labour market conditions were more pronounced in previous generations.

We also conducted the Income Factor GWAS for the male and female results separately and found that their genetic correlation was statistically indistinguishable from 1 ($r_g$ = 0.98, s.e. = 0.02).

## Comparison with EA
### Genetic correlation with EA
To compare the GWAS results for the Income Factor with those for EA, we first conducted an auxiliary GWAS on EA to obtain the most-powered GWAS result for EA with the summary statistics currently available to us. We first carried out a GWAS of EA in the UKB, on the basis of the protocol of the latest EA GWAS (EA4)[34]. We then meta-analysed these GWAS results with the EA3 summary statistics[21] that did not include the UKB, using the meta-analysis version of MTAG. While previous GWASs on income found somewhat inconsistent results on the genetic correlation between EA[21,34] and income ($r_g$ = 0.90, s.e. = 0.04 (ref. 23) and $r_g$ = 0.77, s.e. = 0.02 (ref. 24)), with much greater precision, we found a high genetic correlation that is very close to the first reported estimate ($r_g$ = 0.917, s.e. = 0.006). Among the input income measures, the genetic correlation with EA was higher for occupational and parental income ($r_g$ = 0.95 and 0.92; s.e. = 0.01 and 0.05, respectively) and lower for individual and household income ($r_g$ = 0.81 and 0.82; s.e. = 0.07 and 0.01, respectively). Furthermore, 138 of 161 loci for the Income Factor overlapped with those for EA.

The $r_g$ estimate of 0.917 between the Income Factor and EA implies that only $1 - 0.917^2 = {\sim}16\%$ of the genetic variance of the Income Factor would remain once the genetic covariance with EA was statistically removed.

### GWAS-by-subtraction
We employed the GWAS-by-subtraction approach using genomic structural equation modelling[29] to identify this residual genetic signal (referred to as 'NonEA-Income'). We identified one locus of genome-wide significance for NonEA-Income, marked by the lead SNP rs34177108 on chromosome 16 (Extended Data Fig. 3c). This locus was previously found to be associated with vitamin D levels and cancer, as well as hair- and skin-related traits such as colour and sun exposure, possibly picking up on uncontrolled population stratification (Supplementary Tables 38–41).

## Polygenic prediction
We conducted polygenic index (PGI) analyses with individuals of European ancestry in the Swedish Twin Registry (STR), which was not included in our meta-analysis. We chose the STR as the main prediction

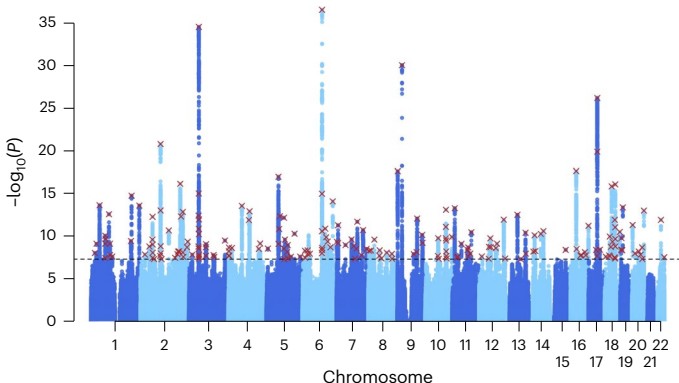

**Fig. 2 | Multivariate GWAS of income.** Manhattan plot presenting the GWAS results for the Income Factor. Unadjusted two-sided Z-test. P values are plotted on the $-\log_{10}$ scale. The red crosses indicate the lead SNPs found from FUMA ($r^2$ < 0.1). The horizontal dashed line indicates genome-wide significance ($P < 5 \times 10^{-8}$).

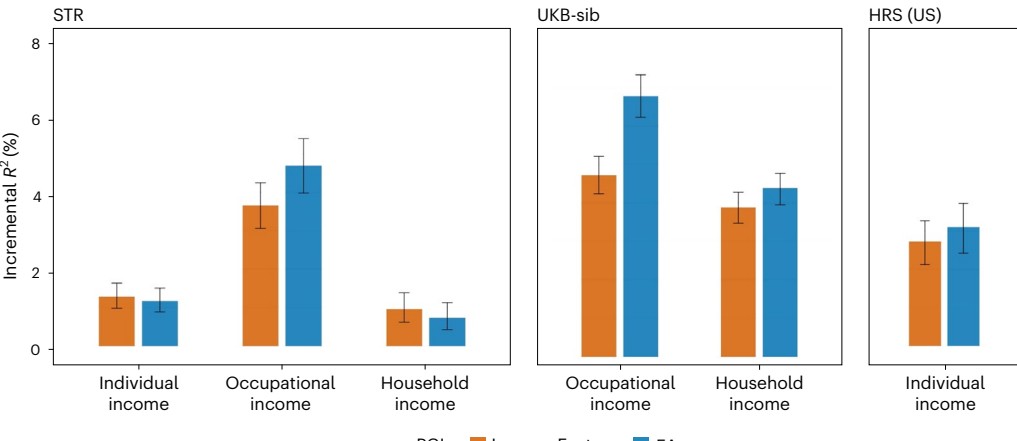

**Fig. 3 | Polygenic prediction of income measures.** Polygenic prediction results in the STR, the UKB-sib and the HRS with PGIs for Income Factor and EA. Prior to fitting the regressions, each phenotype was residualized for demographic covariates (sex, a third-degree polynomial of age and interactions with sex) within each wave, and the mean of the residuals was obtained across the waves for each individual (only a single wave for the UKB-sib). Incremental $R^2$ is the difference between the $R^2$ from regressing the residualized outcome on the PGI and the controls (20 genetic PCs and genotyping batch indicators) and the $R^2$ from a regression only on the controls. Only individuals of European ancestry were included, and one sibling from each family was randomly chosen: $N = 24,946$ (individual), 19,245 (occupational) and 15,655 (household) for the STR; 15,556 (occupational) and 18,303 (household) for the UKB-sib; and 6,171 (individual) for the HRS. The error bars indicate 95% CIs obtained by bootstrapping the sample 1,000 times.

cohort because it has twins and administrative data on individual, occupational and household income. We also used the UKB siblings (UKB-sib) and the Health and Retirement Study (HRS) from the USA as prediction cohorts. For the UKB-sib, occupational and household income measures were available. For the HRS, a self-reported individual income measure was available. In the STR and the UKB-sib cohorts, except when examining within-family prediction, we randomly selected only one individual from each family.

After generating hold-out versions of GWASs on the Income Factor and EA to remove the sample overlap with each prediction sample, we constructed PGIs for the Income Factor and EA using LDpred2 (ref. 35). Before conducting prediction analyses, we residualized the log of income on demographic covariates, including a third-degree polynomial of age, the year of observation and interactions with sex. We measured the prediction accuracy as the incremental $R^2$ from adding the PGI to a regression of the phenotype on a set of baseline covariates, which were the top 20 genetic PCs and genotype batch indicators.

A cohort-specific upper bound for the theoretically possible predictive accuracy of PGIs on income can be obtained by the GREML[36] estimate of the SNP-based heritability of income, which is close to 10% for the available income measures in the STR and UKB-sib samples (Supplementary Table 13). The actual predictive accuracy of PGIs for income is lower than the theoretical maximum, primarily due to finite GWAS sample size but also due to imperfect genetic correlations across meta-analysed cohorts and differences in measurement accuracy of income across samples[37].

In the STR (Fig. 3), the Income Factor PGI predicted $\Delta R^2 = 1.3\%$ (95% CI, 1.0–1.6%) for individual income, 3.7% (95% CI, 3.1–4.2%) for occupational income and 1.0% (95% CI, 0.6–1.4%) for household income. The EA PGI had predictive accuracy results in a similar range for individual and household income, but not for occupational income, for which the accuracy was larger: $\Delta R^2 = 4.7\%$ (95% CI, 4.0–5.4%). Supplementary Fig. 2 shows average income levels per PGI quintile in the STR sample. The expected income of individuals increases monotonically for higher PGI quintiles. Predictive accuracy is the highest for individual income, the most accurate measure of income (derived from Swedish registry data). The difference in average income for individuals in the lowest and highest quintiles of the PGI distribution is ~0.2 standard deviations.

In the UKB-sib, the predictive accuracy of the Income Factor PGI was $\Delta R^2 = 4.7\%$ (95% CI, 4.3–5.2%) for occupational income and 3.9% (95% CI, 3.5–4.3%) for household income. The EA PGI achieved a better predictive accuracy for occupational income ($\Delta R^2 = 6.9\%$; 95% CI, 6.3–7.4%) but only slightly a higher one for household income ($\Delta R^2 = 4.4\%$; 95% CI, 3.9–4.8%). In terms of the coefficient estimates in the UKB-sib, a one-standard-deviation increase in the Income Factor PGI was associated with a 7.2% increase in occupational income (95% CI, 6.7–7.7%) and a 12.3% increase in household income (95% CI, 11.4–13.2%). These estimates are comparable to the effect of one additional year of schooling on income, whose estimates tend to range from 5% to 15% (refs. 8,9,38).

In the HRS, the Income Factor PGI had $\Delta R^2 = 2.7\%$ (95% CI, 2.1–3.3%) for predicting individual income, which was close to the EA PGI's result ($\Delta R^2 = 3.1\%$; 95% CI, 2.4–3.8%).

The predictive power of the Income Factor PGI approached zero once EA or the EA PGI was controlled for. In the UKB-sib, $\Delta R^2$ decreased below 1% for occupational and household income, while the estimates were still statistically different from zero (Extended Data Fig. 4 and Supplementary Table 21).

Although the income PGI is useful for population-level analyses, its predictive accuracy is far too low to make forecasts about the income of any specific individual (Supplementary Information FAQ Section 3.2). Furthermore, the predictive accuracy of our income PGI is substantially reduced from 4–5% among European-ancestry samples to 0–2% among African, Caribbean, Indian, East Asian and South Asian samples in the UKB (Supplementary Fig. 3 and Supplementary Information Section 5.3).

## Direct versus indirect genetic effects

We estimated the share of the direct genetic effect in the overall population effect captured by the Income Factor PGI, following the recent approach that imputes parental genotypes from first-degree relatives[34,39]. Using the UKB-sib sample, we isolated the direct effect of the PGI from the population effect on occupational and household income by controlling for parental PGIs. We found that the ratio of direct effect to population effect estimates is 0.51 (s.e. = 0.05) for occupational income and 0.49 (s.e. = 0.05) for household income (Supplementary Table 22). These results imply that only 24.0% or 25.7% of the Income Factor PGI's predictive power was due to direct genetic

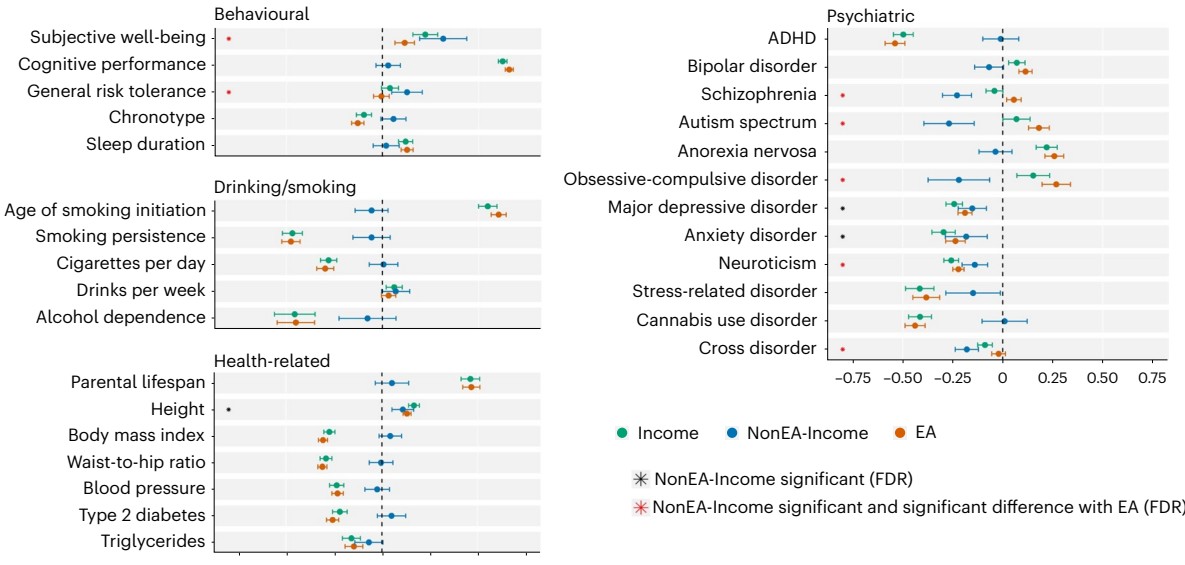

**Fig. 4 | Genetic correlation estimates with health outcomes.** Genetic correlation estimates of Income Factor, NonEA-Income and EA with health outcomes. Point estimates were obtained from LDSC and are displayed as dots. The whiskers show 95% CIs. The black asterisks indicate statistical significance of NonEA-Income at the FDR of 5%. The red asterisks indicate that the estimate

is also significantly different from the estimate for EA at the FDR of 5%. The standard error for the difference was computed from jackknife estimates. Detailed results for all traits, including the sample size for each of the traits, is presented in Supplementary Table 23. ADHD, attention deficit hyperactivity disorder.

effects, which was very close to the result for the EA PGI estimated elsewhere (25.5%)[39].

## Income and health
### Genetic correlations with psychiatric and health traits
We next explored the genetic correlations of the Income Factor, EA and NonEA-Income with phenotypes related to behaviours, psychiatric disorders and physical health (Fig. 4). LDSC estimates revealed that the genetic correlations of EA and the Income Factor largely align. However, noticeable differences emerged for traits in the psychiatric and psychological domains. Specifically, NonEA-Income is associated with a reduced risk for certain psychiatric disorders previously reported to correlate positively with EA[40–42]. These discrepancies were observed for schizophrenia ($r_g = -0.29$, s.e. = 0.04), autism spectrum ($r_g = -0.27$, s.e. = 0.06) and obsessive-compulsive disorder ($r_g = -0.22$, s.e. = 0.08). One possible interpretation of these findings is that these psychiatric disorders have more severe negative effects on individual performance in the labour market than in the educational system.

Intriguingly, NonEA-Income exhibits a near-zero genetic correlation with cognitive performance ($r_g = 0.03$, s.e. = 0.03). At the same time, both EA and the general Income Factor display strong positive genetic correlations with this factor ($r_g = 0.66$, s.e. = 0.01 and $r_g = 0.63$, s.e. = 0.01, respectively). This may suggest that high cognitive performance primarily influences income through education. Furthermore, this result is consistent with high income being attainable through social connections, inherited wealth, entrepreneurial activities or well-paying jobs that do not require high cognitive performance.

While EA and the general Income Factor have substantial negative genetic correlations with health-related behaviours such as age of smoking initiation, smoking persistence, cigarettes per day and alcohol dependence, we found that NonEA-Income has near-zero genetic correlations with these traits (although the latter have substantially larger error margins of the point estimates).

NonEA-Income also displayed genetic correlations with other phenotypes that are similar to those of EA. Specifically, NonEA-Income had negative genetic correlations with major depressive disorder ($r_g = -0.15$, s.e. = 0.04), anxiety disorder ($r_g = -0.19$, s.e. = 0.05) and the

related trait of neuroticism ($r_g = -0.14$, s.e. = 0.03), but positive genetic correlations with subjective well-being ($r_g = 0.32$, s.e. = 0.06), general risk tolerance ($r_g = 0.13$, s.e. = 0.04) and height ($r_g = 0.11$, s.e. = 0.03). The differences in correlations for neuroticism, subjective well-being and risk tolerance were substantial when comparing EA and NonEA-Income, with NonEA-Income showing stronger positive correlations with well-being and risk tolerance and a less negative correlation with neuroticism (Supplementary Table 23).

### Phenome-wide association study on electronic health records
Next, we conducted a phenome-wide association study of the Income Factor PGI on the basis of electronic health records from the UKB-sib's holdout sample. We tested 115 diseases with sex-specific sample prevalence no lower than 1%. In total, 50 diseases from different categories were associated with the Income Factor PGI after Bonferroni correction and 14 after controlling for parental PGI (Fig. 5, Extended Data Fig. 5 and Supplementary Table 27a,b). In all cases, a higher Income Factor PGI value was associated with reduced disease risk, including reduced risk for hypertension, gastroesophageal reflux disease, type 2 diabetes, obesity, osteoarthritis, back pain and depression. The strongest association of a higher Income Factor PGI and a disease was found for essential hypertension.

## Biological annotation
We used functional mapping and annotation of genetic associations (FUMA)[43] to find genes implicated in the Income Factor GWAS. FUMA uses four mapping approaches: positional, chromatin interaction, expression quantitative trait locus mapping and MAGMA gene-based association tests. In total, 2,385 protein-coding genes were implicated by at least one of the methods, of which 225 genes were implicated by all four methods (Extended Data Fig. 6a). Only three of these commonly implicated genes were unique for the Income Factor, compared with the genes implicated in EA GWASs by at least one of the four methods or previously prioritized for EA[21].

We then performed tissue-specific enrichment analyses using LDSC-SEG[44] and MAGMA gene-property analyses[45] (Supplementary Information Section 7). Both methods indicated dominant enrichment

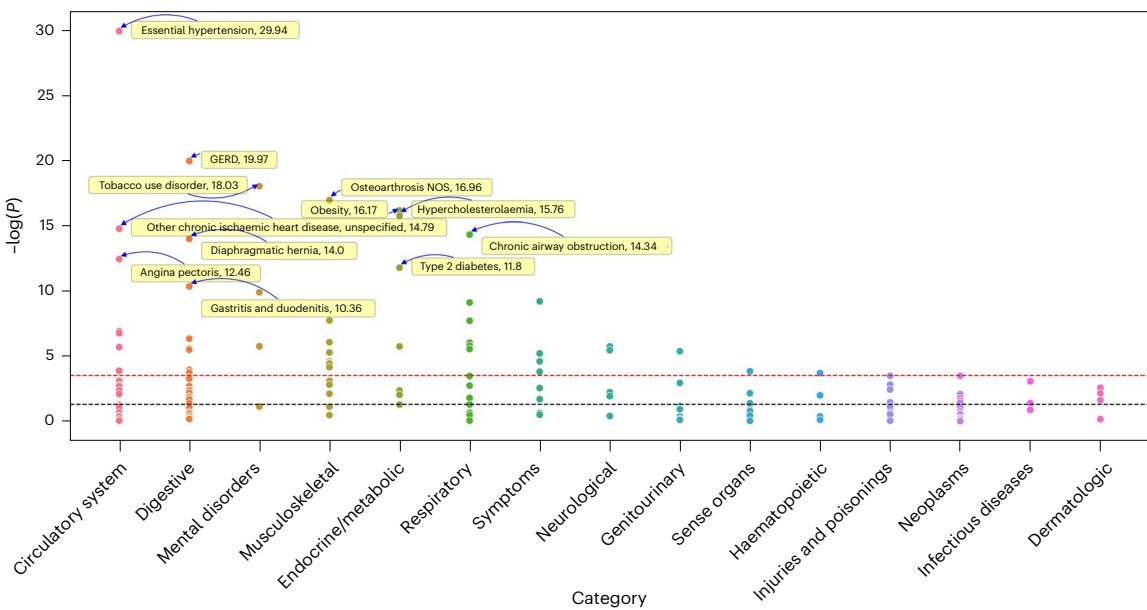

**Fig. 5 | Phenome-wide association study of the Income Factor PGI (without parental PGI controls) in electronic health records for the UKB-sib sample.** The genetic association of Income Factor PGI with 115 diseases from 15 categories without controlling for parental PGIs. The yellow boxes, with arrows pointing to the observations and −log₁₀(P) values reported after the phenotypes, highlight diseases that are strongly associated with the Income Factor PGI (−log₁₀($P$) > 10). The $P$ values were obtained via unadjusted two-sided $Z$-tests. The black and red dashed lines represent the threshold for statistical significance at $P < 0.05$. GERD, gastroesophageal reflux disease.

for tissues of the central nervous system (Extended Data Fig. 6b), consistent with the previous results for household income and EA[21,24].

Next, we compared the genes identified with MAGMA for the Income Factor with those identified for EA and household income. We found that of the 368 genes associated with the Income Factor, 98 had not yet been discovered for EA or household income (Extended Data Fig. 7a and Supplementary Tables 32–34). We further examined the biological processes of genes associated with the Income Factor, EA and household income with FUMA GENE2FUNC. Using a test of overrepresentation, we found three biological processes at a false discovery rate (FDR) of <0.05 that are unique to the Income Factor: neuronal migration (FDR = 0.012), bone formation in early development (FDR = 0.036) and the formation of axons (FDR = 0.047). The overlap among biological processes detected for each trait at FDR < 0.05 is shown in Extended Data Fig. 7b (Supplementary Tables 35–37).

## Discussion

We conducted the largest GWAS on income to date, incorporating individual, household, occupational and parental income measures. Our study design provided increased statistical power, identifying more genetic variants and improving the predictive power of the PGI compared with previous income GWASs. It also allowed for comprehensive additional analyses. Furthermore, we found a strong genetic correlation between income and EA.

Our analyses highlight numerous associations between better health and higher income that are influenced by genetic differences among individuals. These better health outcomes include lower body mass index, blood pressure, type 2 diabetes, depression and stress-related disorders. We note that the genetic overlap between income and health could be driven by different causal mechanisms, including pleiotropic effects of genes, limited income opportunities for individuals with health problems or health advantages for individuals with higher income. Investigating these causal mechanisms is outside the scope of this study.

Previous work examining the relationship between different measures of SES have found that household income, EA, occupational prestige and social deprivation all draw on similar underlying heritable traits[46]. Despite this general genetic factor of SES, our study demonstrates that trait-specific loci are also evident, indicating that income and EA capture heritable traits unique to each of them. Specifically, we estimate that 16% of the genetic variance in income is not shared with EA. The relevance of these income-specific genetic effects is underscored by several diverging relationships with health outcomes between EA and the genetic components of income not shared with EA (NonEA-Income). For example, the genetic correlation with schizophrenia differs between income and EA (income and schizophrenia: $r_g = -0.04$, s.e. = 0.02; EA and schizophrenia: $r_g = 0.06$, s.e. = 0.02; Supplementary Table 23). This divergence is even stronger when NonEA-Income is considered (schizophrenia and NonEA-Income: $r_g = -0.23$, s.e. = 0.04). Furthermore, we found negative genetic correlations of NonEA-Income with bipolar disorder, autism and obsessive-compulsive disorder, while EA exhibits positive genetic correlations with these psychiatric outcomes. This may indicate that the educational system is more accommodating to individuals with these disorders than the labour market and/or that talents associated with these genetic risks (for example, higher IQ with autism[47] or creativity with bipolar disorder and schizophrenia[48]) are more advantageous in school than in the labour market.

More generally, the genetic components of the NonEA-Income factor showed weaker associations with physical health and health-related behaviour, such as drinking and smoking, than those of EA. One possible interpretation of this finding is that better health outcomes of higher SES in wealthy countries could be due more to their association with education than with income or wealth, consistent with findings from quasi-experimental studies[47–49].

While our GWAS results contribute to constructing an income-specific PGI with improved predictive accuracy, the EA PGI remains a comparable or even better predictor of income and SES. This is due to even larger sample sizes in recent GWASs on EA ($N \approx 3,000,000$), lower measurement error in EA than in measures of income and the high genetic correlation between income and EA.

It is important to point out that the results of our study reflect the specific social realities of the analysed samples and are not universal or unchangeable. This is exemplified by the substantial heterogeneity

in the genetic architecture of income that we found across our cohorts of European descent, as well as the non-perfect genetic correlation between sexes. This heterogeneity is consistent with previous findings where the polygenic signal for other measures of SES (such as EA) varies by culture[20] and by country[50]. This genetic heterogeneity is indicative of phenotypic heterogeneity between cultures, where the heritable traits linked to income may not be universal but rather vary and reflect the differences between societies in which heritable traits are facilitative of income differences.

We emphasize that our results are limited to individuals whose genotypes are genetically most similar to the EUR panel of the 1000 Genomes reference panel compared with people sampled in other parts of the world. Our results have limited generalizability and do not warrant meaningful comparisons across different groups or predictions of income for specific individuals (see FAQ in the Supplementary Information). To increase the representation of individuals from diverse backgrounds, cohort and longitudinal studies should seek to sample more diverse and representative samples of the global population.

Our results contribute to the understanding of genetic and environmental factors that influence income. Future research could focus on disentangling these relationships further by integrating genomic data with longitudinal assessments of environmental exposures and behavioural traits. Such approaches could help elucidate the pathways through which genetic predispositions interact with socio-economic contexts, life experiences and individual behaviours to shape income-related outcomes. This line of research may ultimately contribute to a deeper understanding of the mechanisms underlying social mobility and economic inequality.

Studies of genetic analyses of behavioural phenotypes have been prone to misinterpretation, such as characterizing identified associated variants as 'genes for income'. Our study illustrates that such characterization is incorrect for many reasons. The effect of each individual SNP on income is minimal, capturing less than 0.01% of the overall variance in income. Furthermore, the genetic loci we identified correlate with many other traits, including education and a wide range of health outcomes. Finally, the finding that only one quarter of the genetic associations we identified are due to direct genetic effects suggests the potential importance of family-specific factors (including potential resemblance between parents) and environmental factors as important drivers of income inequality.

## Methods
This section provides an overall summary of the analysis methods. Further details are available in the Supplementary Information.

### GWAS meta-analysis
We preregistered our analysis plan for the main income GWAS meta-analysis on 30 August 2018 (https://osf.io/rg8sh/). We used four measures of income (individual, occupational, household and parental income) and conducted a multivariate GWAS to combine these different measures. In total, we recruited 32 cohorts. Some of these cohorts contributed to multiple income measures. Supplementary Tables 1 and 2 summarize the income measures used for each cohort. Supplementary Information Section 2.1 provides details on the phenotype definition. The study was limited to 1KG-EUR-like individuals who were not enrolled in an educational programme at the time of survey or who were above the age of 30 if their current enrolment status was unknown.

Each cohort conducted the additive association analysis as follows. The log-transformed income measure was regressed on the count of effect-coded alleles of the given SNP, controlling for any sources of variation in income that do not reflect individual earning potential according to the data availability of each cohort. This included hours worked (with square and cubic terms), year of survey,

indicators of employment status (such as retired or unemployed), self-employment and pension benefits (Supplementary Table 4). In addition, the covariates included at least the top 15 genetic PCs and cohort-specific technical covariates related to genotyping (genotyping batches and platforms). This analysis was performed for male and female samples separately.

We applied a stringent quality-control protocol based on the EasyQC software package[51] to the GWAS results from each cohort (see Supplementary Information Section 2.4 for more detail). To combine multiple GWAS results on different income measures collected from multiple cohorts, we performed the meta-analysis in several steps. First, for each income measure and each sex, we meta-analysed the cohort-level GWAS results with METAL[27] using sample-size weighting. Then, for each income measure, we meta-analysed the male and female results by using the meta-analysis version of MTAG[28]. To extract the common genetic factor from the four GWAS results with different income measures, we again leveraged MTAG, allowing for different heritabilities among the input traits.

Independent loci were identified using FUMA[42]. First, independent significant SNPs were defined using a cut-off of $P < 5 \times 10^{-8}$ and as independent from any other SNP ($r^2 < 0.6$) within a 1-Mb window. Next, lead SNPs were identified as significant SNPs independent from each other at $r^2 < 0.1$. Finally, independent genomic loci were formed from all independent signals that were in physical proximity to each other by merging independent significant SNPs closer than 250 kb into a single locus using the 1000 Genomes EUR reference panel to ensure that the accuracy of the loci borders was not influenced by missing data in our GWAS. The distance between two loci defined by FUMA is thus between the SNPs in linkage disequilibrium with the independent significant SNPs rather than between the independent significant SNPs themselves.

### Cross-sex and cross-country heterogeneity
We investigated the potential environmental heterogeneity in the GWAS of income by estimating the cross-cohort genetic correlations by sex or by country with LDSC[39]. Sex-specific meta-analysis results for each income measure were available as intermediary outputs from the meta-analysis procedure. In addition, we conducted an Income Factor GWAS on the sex-specific results, which yielded an effective sample size of 360,197 for men and 353,429 for women.

To derive country-specific GWAS meta-analyses, we used only occupational and household income, for which we were able to obtain a sufficiently large sample size for multiple countries. We obtained the household income GWAS for the USA ($N_{eff}$ = 30,855), the UK ($N_{eff}$ = 387,579) and the Netherlands ($N_{eff}$ = 40,533); and the occupational income GWAS for Estonia ($N_{eff}$ = 75,682), Norway ($N_{eff}$ = 42,204), the UK ($N_{eff}$ = 279,883) and the Netherlands ($N_{eff}$ = 24,425).

### Comparative analysis with EA
We compared our Income Factor GWAS results with the GWAS of EA by examining genetic correlation with LDSC and using the GWAS-by-subtraction approach[52]. Here we used a version of EA summary statistics slightly different from publicly available ones. The latest EA GWAS revised the coding of the years of schooling in the UKB[33] to better reflect the educational qualifications of the participants. On the basis of the new coding, we conducted a GWAS of EA in the UKB. Then, by using MTAG with the meta-analysis option, we meta-analysed the UKB result with EA3 summary statistics[21] that did not include the UKB.

We then statistically decomposed the estimated genetic association of the Income Factor into the indirect effect due to EA and the direct effect unexplained by EA (NonEA-Income), leveraging the GWAS-by-subtraction approach in genomic structural equation modelling[34,52]. We implemented this method in the form of a mediation model.

## PGI analysis

We conducted three sets of analyses based on the PGI: (1) prediction analysis, (2) direct genetic effect estimation and (3) a phenome-wide association study of common diseases.

For the PGI prediction analysis, we used the STR[53], the UKB-sib and the HRS[54]. We constructed PGIs using the meta-analysis results for income excluding one prediction cohort at a time, as well as a PGI based on the EA GWAS summary statistics constructed in the same way for comparison. PGIs were created only with HapMap 3 SNPs[55], as these SNPs have good imputation quality and provide good coverage for 1KG-EUR-like individuals. We derived PGIs on the basis of a Bayesian approach implemented in the software LDpred2 (ref. 29).

We measured the prediction accuracy on the basis of incremental $R^2$, which is the difference between the $R^2$ from a regression of the phenotype on the PGI and the baseline covariates and the $R^2$ from a regression on the baseline covariates only. Because income typically contains substantial demographic variation, we pre-residualized the log of income for demographic covariates. Then, as baseline covariates, we included only the top 20 genetic PCs and genotype batch indicators. Because income data were available for multiple years for the STR and the HRS, we residualized the log of income for age, $age^2$, $age^3$, sex, and interactions between sex and the age terms within each year and obtained the mean of residuals for each individual. For the UKB-sib, which had only cross-sectional data, we residualized the log of income for age, $age^2$, $age^3$, sex, dummies for survey year, and interactions between sex and the rest. For the EA measure (years of education), we applied the same procedure with birth-year dummies. We constructed CIs for the incremental $R^2$ by bootstrapping the sample 1,000 times.

To estimate the direct genetic effect of the Income Factor PGI, we used snipar[38] to impute missing parental genotypes from sibling and parent–offspring pairs. Parental PGIs were then created with the imputed SNPs. We estimated the direct genetic effect of the PGI by controlling for the parental PGI. This analysis was conducted only with the UKB-sib sample. See Supplementary Information Section 5.2 for further details.

To explore the clinical relevance of the Income Factor PGI for common diseases, we carried out a phenome-wide association study, using the in-patient electronic health records for 115 diseases with sex-specific sample prevalence no lower than 1% in the UKB-sib sample. We derived case–control status according to the phecode scheme by mapping the UKB's ICD-9/10 records to phecodes v.1.2 (ref. 56). We fitted a linear regression of case–control status on the Income Factor PGI while controlling for the parental PGIs to capture the direct genetic effects of income PGI. As covariates, we also included the year of birth, its square term and its interactions with sex, genotype batch dummies and 20 genetic PCs. Standard errors were clustered by family.

### Reporting summary

Further information on research design is available in the Nature Portfolio Reporting Summary linked to this article.

## Data availability

The GWAS summary statistics are available at https://doi.org/10.62891/aac85602. The data for our analyses come from many cohorts and organizations, some of which are subject to a material transfer agreement, and are listed in the Supplementary Information and Supplementary Table 1. Individual-level data are subject to privacy restrictions and can be requested directly from the participating cohorts.

## Code availability

The code is available at https://doi.org/10.62891/aac85602.

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

## Acknowledgements

This research was conducted using the UKB resource under application no. 11425. We acknowledge the STR for access to data. The STR is managed by the Karolinska Institutet and receives funding through the Swedish Research Council under grant no. 2017-00641. This study was supported by funding from an ERC Consolidator Grant (no. 647648 EdGe, P.D.K.). The funders had no role in study design, data collection and analysis, decision to publish or preparation of the manuscript. Data collection, genotyping, quality control and imputation of genetic data in the Estonian Biobank were carried out by A. Metspalu, L. Milani, R. Mägi, M. Nelis and G. Hudjashov. A full list of acknowledgments is provided in the Supplementary Information.

## Author contributions

H.K., C.A.P.B., T.A.D., R.K.L., A.O., R.d.V. and P.D.K. designed the GWAS meta-analysis. P.D.K. oversaw the study. C.A.P.B. was the lead analyst for the meta-analysis and was responsible for the GWAS, quality control and meta-analysis. H.K. was the lead analyst for the follow-up analyses, including heterogeneity, MiXeR, GWAS-by-subtraction, genetic correlation, PGI prediction and biological annotation analyses. Y.N. performed the cohort-level genetic correlation analysis and polygenic prediction analysis on non-European ancestry groups, and assisted with the follow-up analyses. R.A. conducted the PGI prediction and heritability analyses in the STR sample. C.X. and W.D.H. contributed to several analyses, including cross-country heterogeneity and biological annotation. H.K., D.J.B. and P.D.K. drafted the manuscript. A.A. wrote Box 1. J.P.B., T.A.D., R.K.L., Q.L., T.T.M., A.O., K.P.H., A.A., W.D.H. and R.d.V. provided important input and feedback on various aspects of the study design and the manuscript. All authors contributed to and critically reviewed the manuscript. The individual contributions of all authors according to the CRediT taxonomy are listed in Supplementary Table 29. W.D.H., and C.X. are supported by a Career Development Award from the Medical Research Council (MRC) [MR/T030852/1] for the project titled "From genetic sequence to phenotypic consequence: genetic and environmental links between cognitive ability, socioeconomic position, and health". For the purpose of open access, the author has applied a 'Creative Commons Attribution (CC BY) licence to any Author Accepted Manuscript version arising from this submission.

## Competing interests

The authors declare no competing interests.

## Additional information

**Extended data** is available for this paper at https://doi.org/10.1038/s41562-024-02080-7.

**Correspondence and requests for materials** should be addressed to Abdel Abdellaoui, W. David Hill or Philipp D. Koellinger.

Hyeokmoon Kweon [1], Casper A. P. Burik [1], Yuchen Ning[1], Rafael Ahlskog[2], Charley Xia [3], Erik Abner [4], Yanchun Bao[5], Laxmi Bhatta[6], Tariq O. Faquih [7], Maud de Feijter[8], Paul Fisher[9], Andrea Gelemanović [10], Alexandros Giannelis [11], Jouke-Jan Hottenga [12], Bita Khalili [13,14], Yunsung Lee [15], Ruifang Li-Gao[7], Jaan Masso[16], Ronny Myhre[17], Teemu Palviainen [18], Cornelius A. Rietveld [19,20], Alexander Teumer [21], Renske M. Verweij[22], Emily A. Willoughby [11], Esben Agerbo [23,24,25], Sven Bergmann [13,14], Dorret I. Boomsma [12,26,27,28], Anders D. Børglum [23,29,30], Ben M. Brumpton [31,32,33], Neil Martin Davies [31,34,35], Tõnu Esko[4], Scott D. Gordon [36], Georg Homuth [37], M. Arfan Ikram [8], Magnus Johannesson [38], Jaakko Kaprio [18], Michael P. Kidd[39,40], Zoltán Kutalik [13,41], Alex S. F. Kwong [42,43], James J. Lee[11], Annemarie I. Luik [8,44], Per Magnus [15], Pedro Marques-Vidal [45,46], Nicholas G. Martin [35], Dennis O. Mook-Kanamori[8,47], Preben Bo Mortensen[23,24,25], Sven Oskarsson [2], Emil M. Pedersen[23,24,25], Ozren Polašek[10,48], Frits R. Rosendaal [7], Melissa C. Smart[9], Harold Snieder [49], Peter J. van der Most [49], Peter Vollenweider[44,45], Henry Völzke[50], Gonneke Willemsen[12,51], Jonathan P. Beauchamp[52], Thomas A. DiPrete [53], Richard Karlsson Linnér [1,54], Qiongshi Lu [55], Tim T. Morris[56], Aysu Okbay [1], K. Paige Harden [57], Abdel Abdellaoui [58]✉, W. David Hill [3,59]✉, Ronald de Vlaming [60], Daniel J. Benjamin [61,62,63] & Philipp D. Koellinger [1,64]✉

[1]Department of Economics, School of Business and Economics, Vrije Universiteit Amsterdam, Amsterdam, the Netherlands. [2]Department of Government, Uppsala University, Uppsala, Sweden. [3]Department of Psychology, School of Philosophy, Psychology and Language Sciences, University of Edinburgh, Edinburgh, UK. [4]Institute of Genomics, University of Tartu, Tartu, Estonia. [5]School of Mathematics, Statistics and Actuarial Sciences, University of Essex, Essex, UK. [6]HUNT Center for Molecular and Clinical Epidemiology, Department of Public Health and Nursing, Norwegian University of Science and Technology, Trondheim, Norway. [7]Department of Clinical Epidemiology, Leiden University Medical Center, Leiden, the Netherlands. [8]Department of Epidemiology, Erasmus MC University Medical Center, Rotterdam, the Netherlands. [9]Institute for Social and Economic Research, University of Essex, Essex, UK. [10]Department of Public Health, University of Split School of Medicine, Split, Croatia. [11]Department of Psychology, University of Minnesota Twin Cities, Minneapolis, USA. [12]Department of Biological Psychology, Vrije Universiteit Amsterdam, Amsterdam, the Netherlands. [13]Department of Computational Biology, University of Lausanne, Lausanne, Switzerland. [14]Swiss Institute of Bioinformatics, Lausanne, Switzerland. [15]Centre for Fertility and Health, Norwegian Institute of Public Health, Oslo, Norway. [16]School of Economics and Business Administration, University of Tartu, Tartu, Estonia. [17]Department of Genetics and Bioinformatics, Norwegian Institute of Public Health, Oslo, Norway. [18]Institute for Molecular Medicine Finland, University of Helsinki, Helsinki, Finland. [19]Department of Applied Economics, Erasmus School of Economics, Erasmus University Rotterdam, Rotterdam, the Netherlands. [20]Rotterdam Institute for Behavior and Biology, Erasmus University Rotterdam, Rotterdam, the Netherlands. [21]Department of Psychiatry and Psychotherapy, University Medicine Greifswald, Greifswald, Germany. [22]Department of Public Administration and Sociology, Erasmus University Rotterdam, Rotterdam, the Netherlands. [23]iPSYCH—the Lundbeck Foundation Initiative for Integrative Psychiatric Research, Aarhus University, Aarhus, Denmark. [24]National Centre for Register-Based Research, Aarhus University, Aarhus, Denmark. [25]School of Business and Social Sciences, Aarhus University, Aarhus, Denmark. [26]Amsterdam Public Health, Amsterdam UMC, Amsterdam, the Netherlands. [27]Amsterdam Reproduction & Development, Amsterdam UMC, Amsterdam, the Netherlands. [28]Complex Trait Genetics, Center for Neurogenomics and Cognitive Research, Vrije Universiteit Amsterdam, Amsterdam, the Netherlands. [29]Department of Biomedicine, Aarhus University, Aarhus, Denmark. [30]Center for Genome Analysis and Personalized Medicine, Aarhus, Denmark. [31]K.G. Jebsen Center for Genetic Epidemiology, Department of Public Health and Nursing, Norwegian University of Science and Technology, Trondheim, Norway. [32]HUNT Center for Molecular and Clinical Epidemiology, Department of Public Health and Nursing, Norwegian University of Science and Technology, Levanger, Norway. [33]Clinic of Medicine, St. Olavs Hospital, Trondheim University Hospital, Trondheim, Norway. [34]Division of Psychiatry and Department of Statistical Sciences, University College London, London, UK. [35]Medical Research Council Integrative Epidemiology Unit, University of Bristol, Bristol, UK. [36]Genetic Epidemiology Lab, Queensland Institute of Medical Research, Brisbane, Queensland, Australia. [37]Interfaculty Institute for Genetics and Functional Genomics, University Medicine Greifswald, Greifswald, Germany. [38]Department of Economics, Stockholm School of Economics, Stockholm, Sweden. [39]Economics, RMIT University, Melbourne, Victoria, Australia. [40]International School

of Technology and Management, Feng Chia University, Taichung, Taiwan. [41]University Center for Primary Care and Public Health, Unisante, Lausanne, Switzerland. [42]MRC Integrative Epidemiology Unit, University of Bristol, Bristol, UK. [43]Division of Psychiatry, University of Edinburgh, Edinburgh, UK. [44]Trimbos Institute—Netherlands Institute for Mental Health and Addiction, Utrecht, the Netherlands. [45]Department of Medicine, Internal Medicine, Lausanne University Hospital (CHUV), Lausanne, Switzerland. [46]Faculty of Biology and Medicine, University of Lausanne, Lausanne, Switzerland. [47]Department of Public Health and Primary Care, Leiden University Medical Center, Leiden, the Netherlands. [48]Algebra University, Zagreb, Croatia. [49]Department of Epidemiology, University of Groningen and University Medical Center Groningen, Groningen, the Netherlands. [50]Institute for Community Medicine, University Medicine Greifswald, Greifswald, Germany. [51]Faculty of Health, Sports and Wellbeing, Inholland University of Applied Sciences, Haarlem, the Netherlands. [52]Interdisciplinary Center for Economic Science and Department of Economics, George Mason University, Fairfax, VA, USA. [53]Department of Sociology, Columbia University, New York, NY, USA. [54]Department of Economics, Leiden Law School, Universiteit Leiden, Leiden, the Netherlands. [55]Department of Biostatistics and Medical Informatics, University of Wisconsin–Madison, Madison, WI, USA. [56]Centre for Longitudinal Studies, Social Research Institute, University College London, London, UK. [57]Department of Psychology and Population Reseach Center, University of Texas at Austin, Austin, TX, USA. [58]Department of Psychiatry, Amsterdam UMC, University of Amsterdam, Amsterdam, the Netherlands. [59]Lothian Birth Cohort Studies, University of Edinburgh, Edinburgh, UK. [60]Department of Econometrics and Data Science, School of Business and Economics, Vrije Universiteit Amsterdam, Amsterdam, the Netherlands. [61]Anderson School of Management, University of California, Los Angeles, Los Angeles, CA, USA. [62]Human Genetics Department, UCLA David Geffen School of Medicine, Los Angeles, CA, USA. [63]National Bureau of Economic Research, Cambridge, MA, USA. [64]DeSci Foundation, Geneva, Switzerland. ✉e-mail: a.abdellaoui@amsterdamumc.nl; David.Hill@ed.ac.uk; p.d.koellinger@vu.nl

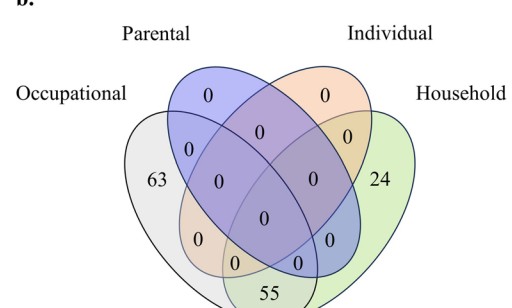

**a.**

**b.**

**Extended Data Fig. 1 | Venn diagram of loci across phenotypes.**
The diagram shows how genome-wide significant loci and genes mapped to the 86 independent loci are distributed across the four income phenotypes. (**a**.) The 86 genome-wide significant loci and their overlap across the four income phenotypes is shown (**b**.) Gene-based statistics were derived using MAGMA for genes whose physical boundaries overlapped with a genome-wide significant loci from the four income phenotypes.

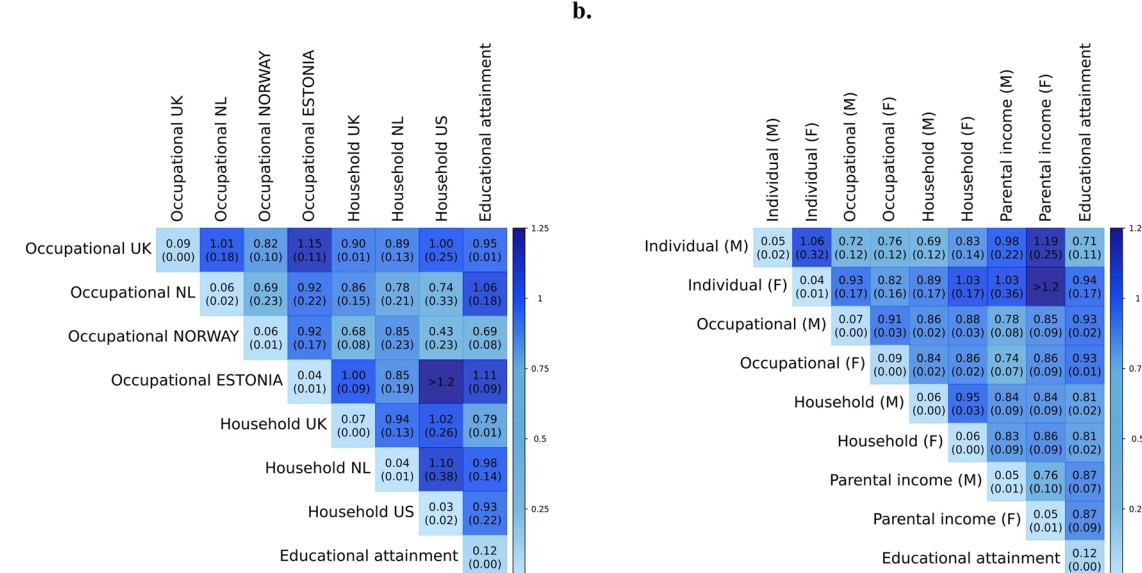

**Extended Data Fig. 2 | Cross-cohort genetic correlations of income stratified by sex and country.** LDSC estimates for cross-cohort genetic correlations of income **(a.)** between countries and **(b.)** between male (M) and female (F). The diagonal elements report SNP heritabilities. The standard errors are reported in parentheses. Some of the results were out-of-bound estimates (exceeding 1).

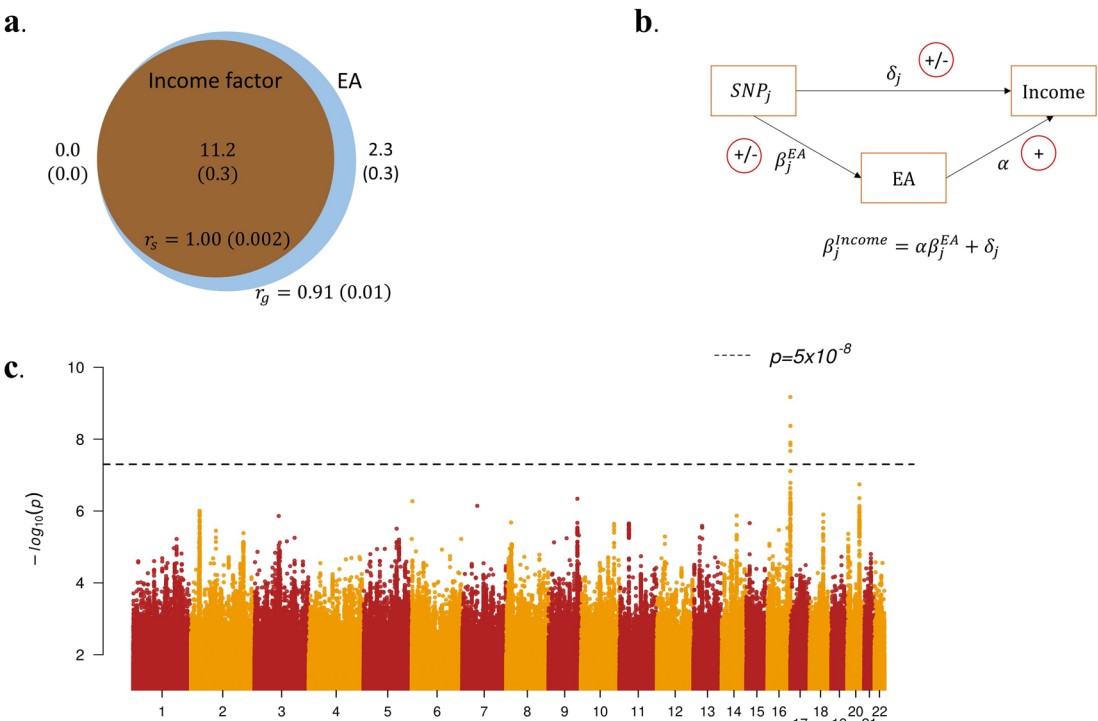

**Extended Data Fig. 3 | Polygenic overlap of income with EA and GWAS-by-subtraction. (a.)** Venn diagram presenting MiXeR results on unique and shared polygenic components for Income Factor (orange) and EA (blue). The estimated numbers of unique and shared variants are represented in thousands and illustrated by the areas of the circles: 0.45 and 2,260 unique variants for income and EA, respectively, and 11,153 shared variants. $r_g$ is the global genetic correlation while $r_s$ is the correlation within the shared variants. The standard errors are reported in the parentheses. **(b.)** The GWAS-by-subtraction model of non-EA income describes the genetic effect of income for SNP $j$ $\left(\beta_j^{INC}\right)$ as the

sum of two components: 1) $\alpha\beta_j^{EA}$: the indirect effect that reflects the genetic association of EA and 2) $\delta_j$: the direct effect of SNPs on income reflects the genetic effect of income after statistically removing its genetic covariance with EA. Note that the diagram only depicts a statistical meditation for interpretation and is not meant to imply any directionality or causal ordering of SNPs to phenotypes. **(c.)** Manhattan plot showing the NonEA genetic associations of the Income Factor (NonEA-Income, corresponding to $\delta_j$ from **b**.). Unadjusted two-sided $Z$-test. $p$-values are plotted on the $-log_{10}$ scale.

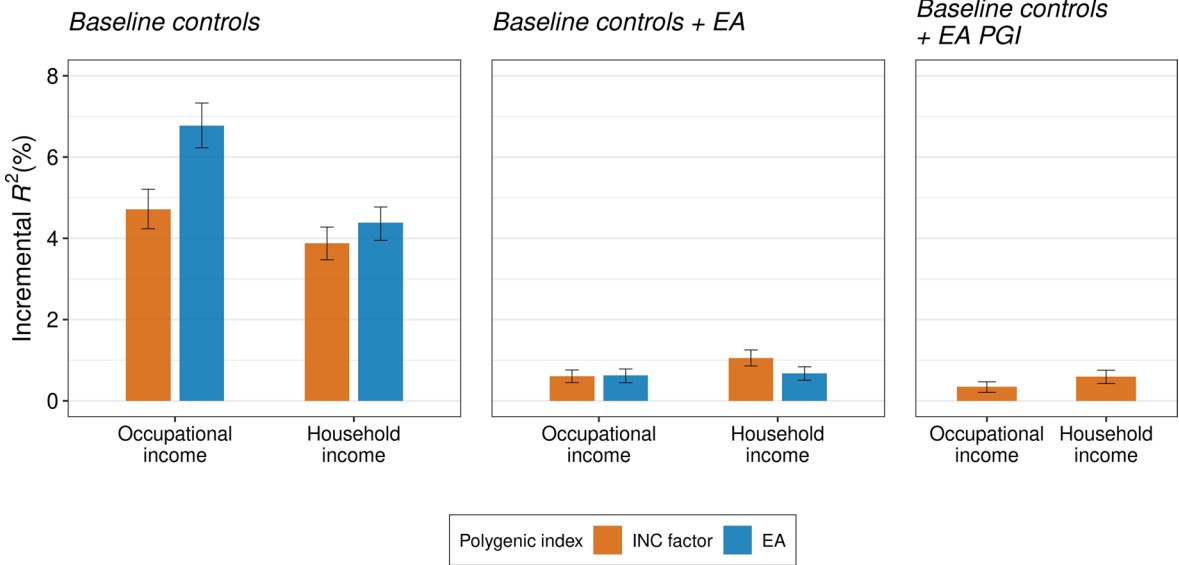

**Extended Data Fig. 4 | Polygenic prediction of income with additional controls.** The figure reports polygenic prediction results in the UKB siblings with the Income Factor PGI and additional controls (EA or the PGI for EA). Before fitting the regressions, each phenotype was residualised for demographic covariates (a third-degree polynomial for age, year of observation, and interactions with sex). The incremental $R^2$ is calculated as the difference between the $R^2$ from regressing the residualised outcome on both the Income Factor PGI and the controls and the $R^2$ from regressing only on the controls. The baseline controls include 20 genetic PCs and genotyping batch indicators. Only individuals of European ancestry were included, and one sibling from each family was randomly chosen. The error bars indicate 95% confidence intervals around the incremental $R^2$ obtained by bootstrapping the sample 1,000 times.

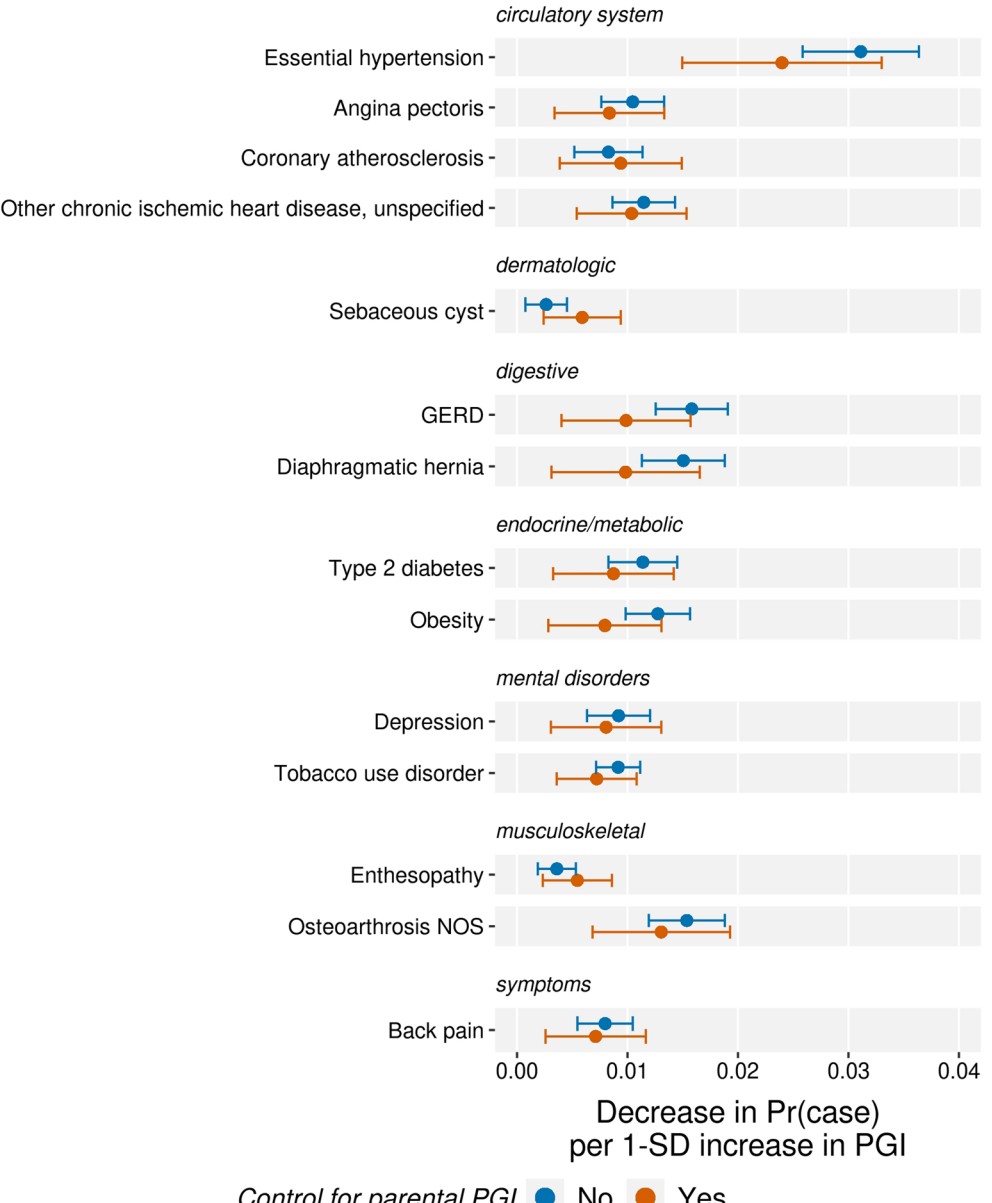

**Extended Data Fig. 5 | Phenome-wide association study of the Income Factor PGI in electronic health records for the UKB sibling sample.** The figure presents results from a phenome-wide association study using in-patient electronic health records from the UKB sibling sample, focusing on 115 diseases with sex-specific prevalence of at least 1%. Case-control status was determined using the phecode v1.2 scheme, which maps the UKB's ICD-9/10 records. The case-control status was regressed on the Income Factor PGI, both with and without controlling for parental PGI. Additional covariates included birth year, its squared term, their interactions with sex, genotype batch dummies, and 20 genetic principal components (PCs). Standard errors were clustered by family. The sign of the coefficient estimates was reversed to reflect a decrease in the probability of having the disease. Results were plotted only for diseases significantly associated with Income Factor PGI at a 5% FDR, with parental PGI controlled for. Dots represent point estimates of the incremental $R^2$ for Income Factor PGI on each disease, while error bars show unadjusted 95% confidence intervals.

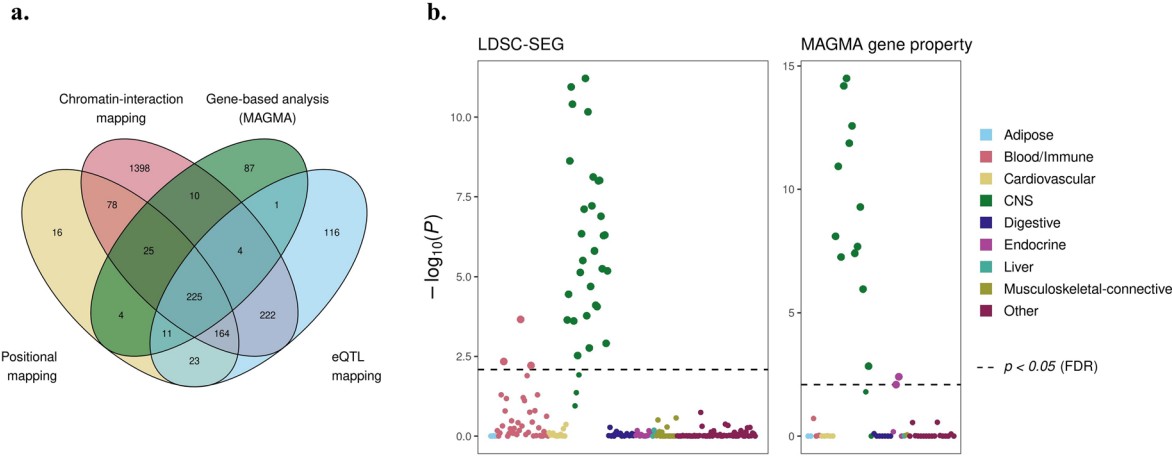

**Extended Data Fig. 6 | Biological annotation. (a.)** The Venn diagram illustrates the overlap of genes implicated in the Income Factor using four methods: positional mapping, eQTL mapping, chromatin interaction mapping, and MAGMA gene-based analysis. **(b.)** The figures show the results of tissue-specific enrichment analysis using LDSC-SEG (left) and MAGMA gene-property analysis (right). Each circle represents a tissue or cell type from the GTEx or Franke lab gene expression datasets, with larger circles indicating statistical significance at a 5% false discovery rate. Full results are available in Supplementary Table 26.

a.

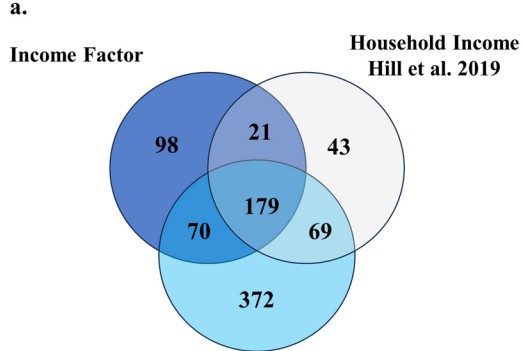

b.

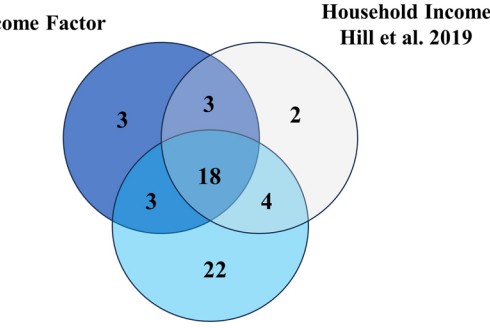

**Extended Data Fig. 7 | Vene diagram of genes associated with the Income Factor, household income, and educational attainment. (a.)** Gene-based statistics for household income and educational attainment were sourced from Hill et al.[24]. and Lee et al.[21]. respectively. A Bonferroni correction was applied for each trait to determine statistical significance. **(b.)** Vene diagram of gene sets associated with the Income Factor, household income and educational attainment based on FUMA GENE2FUNC analyses and a test of overrepresentation at FDR <0.05. See Supplementary Tables 35–37 for further details.

# Reporting Summary

## Statistics

For all statistical analyses, confirm that the following items are present in the figure legend, table legend, main text, or Methods section.

| n/a | Confirmed | |
|---|---|---|
| ☐ | ☒ | The exact sample size (*n*) for each experimental group/condition, given as a discrete number and unit of measurement |
| ☐ | ☒ | A statement on whether measurements were taken from distinct samples or whether the same sample was measured repeatedly |
| ☐ | ☒ | The statistical test(s) used AND whether they are one- or two-sided<br>*Only common tests should be described solely by name; describe more complex techniques in the Methods section.* |
| ☐ | ☒ | A description of all covariates tested |
| ☐ | ☒ | A description of any assumptions or corrections, such as tests of normality and adjustment for multiple comparisons |
| ☐ | ☒ | A full description of the statistical parameters including central tendency (e.g. means) or other basic estimates (e.g. regression coefficient) AND variation (e.g. standard deviation) or associated estimates of uncertainty (e.g. confidence intervals) |
| ☐ | ☒ | For null hypothesis testing, the test statistic (e.g. *F*, *t*, *r*) with confidence intervals, effect sizes, degrees of freedom and *P* value noted<br>*Give P values as exact values whenever suitable.* |
| ☒ | ☐ | For Bayesian analysis, information on the choice of priors and Markov chain Monte Carlo settings |
| ☒ | ☐ | For hierarchical and complex designs, identification of the appropriate level for tests and full reporting of outcomes |
| ☐ | ☒ | Estimates of effect sizes (e.g. Cohen's *d*, Pearson's *r*), indicating how they were calculated |

*Our web collection on statistics for biologists contains articles on many of the points above.*

## Software and code

Policy information about availability of computer code

| | |
|---|---|
| Data collection | N/A (Our study is entirely based on previously existing data) |
| Data analysis | METAL, release 2011-03-25, http://csg.sph.umich.edu/abecasis/metal/; MTAG software v.1.0.1, https://github.com/omeed-maghzian/mtag; GSEM v.0.0.3e https://github.com/GenomicSEM/GenomicSEM/wiki; LDSC and LDSC-SEG v1.0.1 https://github.com/bulik/ldsc; MAGMA v1.10 https://ctg.thebluebus.nl/software/magma; COJO and GREML in GCTA v1.94 https://yanglab.westlake.edu.cn/software/gcta/#Download; LDpred2 https://github.com/privefl/paper-ldpred2/tree/master; FUMA https://fuma.ctglab.nl/tutorial#overview, EasyQC 23.8 https://www.uni-regensburg.de/medizin/epidemiologie-praeventivmedizin/genetische-epidemiologie/software/index.html, snipar https://github.com/AlexTISYoung/snipar |

For manuscripts utilizing custom algorithms or software that are central to the research but not yet described in published literature, software must be made available to editors and reviewers. We strongly encourage code deposition in a community repository (e.g. GitHub). See the Nature Portfolio guidelines for submitting code & software for further information.

## Data

Policy information about availability of data

All manuscripts must include a data availability statement. This statement should provide the following information, where applicable:
- Accession codes, unique identifiers, or web links for publicly available datasets
- A description of any restrictions on data availability
- For clinical datasets or third party data, please ensure that the statement adheres to our policy

GWAS summary statistics are available at https://beta.dpid.org/149. Data for our analyses come from many cohorts and organizations, some of which are subject to a MTA, and are listed in the Supplementary Information and Supplementary Table 1. Individual-level data are subject to privacy restrictions and can be requested directly from the participating cohorts.

## Research involving human participants, their data, or biological material

Policy information about studies with human participants or human data. See also policy information about sex, gender (identity/presentation), and sexual orientation and race, ethnicity and racism.

| Reporting on sex and gender | We conducted sex-stratified GWAS on income in all participating samples. Sex-stratified analyses are clearly described as such in the manuscript and supplementary information. We share both sex-stratified and aggregated GWAS summary statistics publicly. |
|---|---|
| Reporting on race, ethnicity, or other socially relevant groupings | We restricted our analyses to 1000 Genomes EUR-like individuals to maximize statistical power and to minimize bias from unobserved environmental factors that are correlated with differences in minor allele frequencies across ancestry groups. As with previous genetic studies on social outcomes like educational attainment, the findings of this study have limited generalisability across different populations. |
| Population characteristics | We restricted our analyses to 1KG-EUR-like individuals who were not currently enrolled in an educational program or who were aged above 30 if their current enrollment status was unknown. |
| Recruitment | Recruitment protocols varied across participating cohorts and are described in the references provided for each dataset (Supplementary Table 1). |
| Ethics oversight | Each participating cohort signed a collaboration agreement verifying that the responsible Institutional Review Board (IRB) or ethical committee has approved a GWAS of income in that sample. |

Note that full information on the approval of the study protocol must also be provided in the manuscript.

# Field-specific reporting

Please select the one below that is the best fit for your research. If you are not sure, read the appropriate sections before making your selection.

☐ Life sciences   ☒ Behavioural & social sciences   ☐ Ecological, evolutionary & environmental sciences

For a reference copy of the document with all sections, see nature.com/documents/nr-reporting-summary-flat.pdf

# Behavioural & social sciences study design

All studies must disclose on these points even when the disclosure is negative.

| Study description | We conducted sex-stratified GWAS and meta-analyzed results from 32 cohorts across 12 economically advanced countries and three continents, yielding the largest GWAS on income to date with an effective sample size of N = 668,288 (Table 1). |
|---|---|
| Research sample | We restricted our analyses to 1000 Genomes EUR-like individuals to maximize statistical power and to minimize bias from unobserved environmental factors that are correlated with differences in minor allele frequencies across ancestry groups. As with previous genetic studies on social outcomes like educational attainment, the findings of this study have limited generalisability across different populations. |
| Sampling strategy | Sampling strategies varied across participating cohorts and are described in the references provided for each dataset (Supplementary Table 1). |
| Data collection | Data collection strategies varied across participating cohorts and are described in the references provided for each dataset (Supplementary Table 1). |
| Timing | Timing of data collection varied across participating cohorts and are described in the references provided for each dataset (Supplementary Table 1). |
| Data exclusions | We restricted our analyses to 1000 Genomes EUR-like individuals to maximize statistical power and to minimize bias from |

| | |
|---|---|
| Data exclusions | unobserved environmental factors that are correlated with differences in minor allele frequencies across ancestry groups. |
| Non-participation | N/A (secondary data analyses) |
| Randomization | N/A |

# Reporting for specific materials, systems and methods

We require information from authors about some types of materials, experimental systems and methods used in many studies. Here, indicate whether each material, system or method listed is relevant to your study. If you are not sure if a list item applies to your research, read the appropriate section before selecting a response.

### Materials & experimental systems

| n/a | Involved in the study |
|---|---|
| ☒ ☐ | Antibodies |
| ☒ ☐ | Eukaryotic cell lines |
| ☒ ☐ | Palaeontology and archaeology |
| ☒ ☐ | Animals and other organisms |
| ☒ ☐ | Clinical data |
| ☒ ☐ | Dual use research of concern |
| ☒ ☐ | Plants |

### Methods

| n/a | Involved in the study |
|---|---|
| ☒ ☐ | ChIP-seq |
| ☒ ☐ | Flow cytometry |
| ☒ ☐ | MRI-based neuroimaging |

## Plants

| | |
|---|---|
| Seed stocks | N/A |
| Novel plant genotypes | N/A |
| Authentication | N/A |

