## [Peer Review File · Nature Human Behaviour]

Associations between common genetic variants and income provide insights about the socioeconomic health gradient

Corresponding Author: Professor Philipp D Koellinger

Version 0:

Decision Letter:

5th February 2024

Dear Professor Koellinger,

Thank you once again for your manuscript, entitled "Associations between common genetic variants and income provide insights about the socioeconomic health gradient", and for your patience during the peer review process.

Your Article has now been evaluated by 3 referees. You will see from their comments copied below that, although they find your work of potential interest, they have raised quite substantial concerns. In light of these comments, we cannot accept the manuscript for publication, but would be interested in considering a revised version if you are willing and able to fully address reviewer and editorial concerns.

We hope you will find the referees' comments useful as you decide how to proceed. If you wish to submit a substantially revised manuscript, please bear in mind that we will be reluctant to approach the referees again in the absence of major revisions. We are committed to providing a fair and constructive peer-review process. Do not hesitate to contact us if there are specific requests from the reviewers that you believe are technically impossible or unlikely to yield a meaningful outcome.

If you wish to submit a suitably revised manuscript, we would hope to receive it within 4 months. I would be grateful if you could contact us as soon as possible if you foresee difficulties with meeting this target resubmission date.

- Include a "Response to the editors and reviewers" document detailing, point-by-point, how you addressed each editor and referee comment. If no action was taken to address a point, you must provide a compelling argument. When formatting this document, please respond to each reviewer comment individually, including the full text of the reviewer comment verbatim followed by your response to the individual point. This response will be used by the editors to evaluate your revision and sent back to the reviewers along with the revised manuscript.
- Highlight all changes made to your manuscript or provide us with a version that tracks changes.

Link Redacted

Thank you for the opportunity to review your work. Please do not hesitate to contact me if you have any questions or would like to discuss the required revisions further.

[redacted]

REVIEWER COMMENTS:

Reviewer #1:

Remarks to the Author:

This paper conducted the first large-scale GWAS on income among individuals of European descent, which provided an important data source for the downstream post-GWAS analysis. However, there are important issues that need to be addressed:

- 1) Please provide the reason why reference 16 and 17 mentioned in the introduction were cited as I couldn't find specific heritability estimates during verification.
- 2) Although time-specific intercepts were employed, how authors controlled the effects of inflation to ensure comparability between longitudinal data and cross-sectional survey data?
- 3) According to the current results from Supplementary Table 22, I can't deduce how these estimates were developed.
 - a) "These results imply that only 24.0% or 25.7% of the INC factor PGI's predictive power was due to direct genetic effects, which was very close to the result for the EA PGI estimated elsewhere (25.5%).³⁸"
- 4) Please provide the proportion of original parental genotypes, imputed parental genotypes with observed data, and imputed parental genotypes with the frequency (f) of allele 1.
- 5) Page 21, there is a clerical error in the Neff of the GWAS in USA.
- 6) The total results section lacks a clear structure. For instance, in the polygenic prediction, the author has actually covered a substantial amount of content, including estimates of direct effect proportions and PheWAS. These results address different scientific questions, but the author just combined them in a crude way without proper organization. Then, they moved on to genetic correlation, which is more closely related with the previous section (Comparison with educational attainment), both in terms of research methods and content.
- 7) Figures provided by the author were incomplete.
- 8) Authors need to add explanation why the genetic correlations estimated were greater than 1. Did it imply data quality is actually very poor?
- 9) Personally, I don't think the title is accurate. The incremental R² didn't reflect the relationship between polygenic score and income. Authors need to provide a detailed distribution of polygenic scores, e.g. correlations with income or in the form of percentiles, to better capture the "gradient." It's a necessary step before the phenome-wide association study of the INC factor PGI on socioeconomic health.
- 10) This research was pre-registered on August 30 2018. And the study is generally consistent with the protocol. However, the INC factor was not included in the protocol. Moreover, the correlation between the constructed INC factor and the actual income was not explained in the manuscript.

Reviewer #2:

Remarks to the Author:

The authors conducted a GWAS of income on individuals of all European ancestry. Although the study has strength of big sample size, I have several major concerns about the study design and GWAS methodology. Below are my specific comments:

The biggest issue of this study is the bias coming from phenotype (i.e., income) and population ancestry (i.e., European). It is well-known that race/ethnicity is strongly related to economic status (e.g., income). However, this study only investigated the European population, which cannot truly reflect the GWAS signal for income due to the lack of other races/ethnicities.

Although the sample size is huge, most samples are from UK Biobank. This study is a meta-analysis (i.e., not single cohort GWAS), I think it will be nice to have individual GWAS of other ancestries, then meta-analysis with the European ancestry. This will benefit from studying population-specific GWAS signals and comparing GWAS signals across different population ancestry. This is especially important for the income phenotype.

I am not clear why age₃ needs to be adjusted in the model for re-visualization. I have seen age and age₂, but it is very rare to adjust age₃ unless it is well-justified.

After GWAS meta-analysis, it is unclear whether the signals are LD-independent or not. It seems like the authors did not perform a clumping analysis.

In addition, a conditional analysis needs to be performed to identify secondary loci that are conditioned on the primary loci, such as using COJO.

The fine-mapping analysis is also missing from this study, but this is standard for GWAS and has to be done.

The results of functional analysis from this study are only confirmatory to the previous findings (references 21 and 24), and not novel. In addition to central nervous system, any new biological function/tissue/system was identified? The extended figure 5 was cut in half, and not showing these results.

In addition, all analyses were based in silico, it will be much stronger to have more mechanistic data (e.g., in vitro, in vivo) to validate the function of identified loci.

The abbreviation "INC" used in this manuscript is not very helpful because it does not shorten the whole word; instead, it just represents income. So why not just use "income", which is clearer?

Reviewer #3:

Remarks to the Author:

I trust this letter finds you well. I am writing to convey the positive outcome of the review process for your manuscript. The quality of your research, the robustness of your methodology, and the clarity of presentation have all contributed to this positive recommendation.

The study is well-organized, the literature review is comprehensive, and the conclusions drawn are supported by the presented data. While the manuscript is well-documented as it stands, I would like to highlight some minor suggestions for improvement that you may consider addressing.

(1) In the first paragraph of results section, four distinct measures of income were delineated. Subsequently, independent GWAS were performed on each of these measures, leading to the identification of 86 non-overlapping loci. However, the biological functions of these loci remain unclear, and it is currently unknown how many loci are shared among the four income measures.

(2) In the analysis of "Cross-sex and cross-country heterogeneity," were the cohorts included in the meta-analysis exclusively of European ancestry?

(3) Additional evidence is required to substantiate the functional role of rs34177108, which was identified through the NonEA-INC GWAS.

Version 1:

Decision Letter:

Our ref: NATHUMBEHAV-23124307A

30th July 2024

Dear Dr Koellinger,

Thank you for submitting your revised manuscript "Associations between common genetic variants and income provide insights about the socioeconomic health gradient" (NATHUMBEHAV-23124307A). It has now been seen by the original referees and their comments are below. As you can see, the reviewers find that the paper has improved in revision. We will therefore be happy in principle to publish it in Nature Human Behaviour, pending revisions to satisfy the referees' final requests and to comply with our editorial and formatting guidelines.

We are now performing detailed checks on your paper and will send you a checklist detailing our editorial and formatting requirements within two weeks. Please do not upload the final materials and make any revisions until you receive this additional information from us.

[redacted]

Reviewer #1 (Remarks to the Author):

The authors have addressed the majority of previous comments satisfactorily. The improvements in methodology, data presentation, and overall clarity are commendable. I have a minor concern regarding the direct and indirect genetic effects. Although this issue occupies only a small portion of the results, it is worth discussing in the context of the overall conclusions. According to the data provided by the authors, only about 20% of the study subjects have complete nuclear family data, while the rest were imputed. In snipar, not all situations can accurately infer parental genotypes (IBD0). In other instances (when siblings share one allele IBD or both alleles IBD), parental alleles are unobserved and imputed with the frequency of allele 1, f. Therefore, I would recommend that the authors provide information on the quality control of the imputation, such as genotyping error rate, proportion of IBD0, or perform a sensitivity analysis on the nuclear families with complete information.

Overall, I am satisfied with the revisions made. Thank you again for the opportunity to review this manuscript. It has been a valuable learning experience for me.

Reviewer #2 (Remarks to the Author):

Overall, the authors addressed most of my comments. However, several issues still need to be addressed:

1. In the current era of GWAS, it is not always bigger and better. After many years of GWAS, it is more important to study non-European populations than European populations. The authors made a claim that it is out of the scope to study non-Europeans in the current study. This is not true. UKB has many non-European subjects with decent sample size. I still suggest the authors include those subjects for the population specific analysis.

2. Some careless mistakes still present. For example, the authors made a claim that they have changed the wording from "INC factor" to "Income Factor" throughout the manuscript. However, Figures 1 and 2, and many other figures still show "INC factor". This shows that the authors did not pay attention to details.

Reviewer #3 (Remarks to the Author):

I am pleased to inform you that I find it to be of high quality with no further revisions needed.

Reviewer #4 (Remarks to the Author):

The authors conducted a GWAS on income in the individuals of European descent, identifying 162 genomic loci associated with a common genetic factor underlying various income measures. Despite the small effect sizes, these findings offer valuable insights into the complex relationship between genetics, income, and health outcomes. The study's polygenic index captures 1-4% of income variance, highlighting the modest but noteworthy genetic influence on income. Several post-GWAS analyses, including PheWAS, further elucidate the health implications, demonstrating reduced risks for several diseases among individuals with a higher polygenic index for income.

While the study's methodological comprehensive approach are commendable, I have some concerns and suggestions.

Major concern:

Considering the high genetic correlation with previous household income GWAS or educational attainment GWAS, I have concerns about the incremental value of the current study. The study identifies 162 genomic loci associated with income, yet these loci have miniscule effect sizes and explain only few portions of the income variance.

Although the lack of novel signals compared to previous GWASs might seem like a limitation, I believe these findings still contribute to the field. It would be beneficial to emphasize the importance of these findings in the context of existing sociogenomic literature and discuss the implications of these "modest" findings, like how they advance our understanding of the genetic architecture of SES. How do these results compare with other socioeconomic traits?

Specific Comments:

1. The rationale for choosing different meta-analysis methods (MTAG vs. METAL) should be clearly articulated. The decision to use MTAG for sex-stratified analyses, rather than combining sexes from the outset, requires further justification. Clarifying these methodological choices will enhance the transparency and reproducibility of the study.
2. The sentence "the estimated effects at the 5th and 95th percentiles were 0.18 and 0.60%, respectively" is ambiguous. It is unclear whether these percentages refer to the proportion of variance explained or another metric. Providing a more detailed explanation and context for these figures is needed.
3. When translating genetic effects into real dollar amounts (e.g. an annual income increase of \$169), provide the standard deviation for these translations.
4. Further discussing the potential future research directions, such as disentangling the interplay between genetic, behavioral, and environmental factors, will underscore the importance of this work.

Version 2:

Decision Letter:

Dear Prof Koellinger,

We are pleased to inform you that your Article "Associations between common genetic variants and income provide insights about the socioeconomic health gradient", has now been accepted for publication in *Nature Human Behaviour*.

Please note that *Nature Human Behaviour* is a Transformative Journal (TJ). Authors may publish their research with us through the traditional subscription access route or make their paper immediately open access through payment of an article-processing charge (APC). Authors will not be required to make a final decision about access to their article until it has been accepted. [Find out more about Transformative Journals](https://www.springernature.com/gp/open-research/transformative-journals)

Once your manuscript is typeset and you have completed the appropriate grant of rights, you will receive a link to your electronic

proof via email with a request to make any corrections within 48 hours. If, when you receive your proof, you cannot meet this deadline, please inform us at rjsproduction@springernature.com immediately. Once your paper has been scheduled for online publication, the Nature press office will be in touch to confirm the details.

[redacted]

P.S. Click on the following link if you would like to recommend Nature Human Behaviour to your librarian <http://www.nature.com/subscriptions/recommend.html#forms>

** Visit the Springer Nature Editorial and Publishing website at http://editorial-jobs.springernature.com?utm_source=ejp_NHumB_email&utm_medium=ejp_NHumB_email&utm_campaign=ejp_NHumB for more information about our career opportunities. If you have any questions please click [here](mailto:editorial.publishing.jobs@springernature.com).

Open Access This Peer Review File is licensed under a Creative Commons Attribution 4.0 International License, which permits use, sharing, adaptation, distribution and reproduction in any medium or format, as long as you give appropriate credit to the original author(s) and the source, provide a link to the Creative Commons license, and indicate if changes were made. In cases where reviewers are anonymous, credit should be given to 'Anonymous Referee' and the source. The images or other third party material in this Peer Review File are included in the article's Creative Commons license, unless indicated otherwise in a credit line to the material. If material is not included in the article's Creative Commons license and your intended use is not permitted by statutory regulation or exceeds the permitted use, you will need to obtain permission directly from the copyright holder.

Reviewer #1:

Remarks to the Author:

This paper conducted the first large-scale GWAS on income among individuals of European descent, which provided an important data source for the downstream post-GWAS analysis. However, there are important issues that need to be addressed:

1) Please provide the reason why reference 16 and 17 mentioned in the introduction were cited as I couldn't find specific heritability estimates during verification.

Taubman (1976, reference 16) provides heritability estimates of income in Table 3, second column (g^2). Taubman used a more general model to estimate heritability by allowing shared environment correlations to differ among DZ and MZ twins. This led to the slightly unusual format of Table 3. Taubman 1976 is the standard reference for the first estimate of income heritability in the economics literature. Quoting from p. 866-867: "It is not clear how to allocate the covariance term between genetics, family, and other environments (in a standard twin model of heritability). But practically any such allocation would indicate that genetics plus family environment account for 30-55 percent of the total variance (in income)..."

You are correct that Visscher et al. (2008, reference 17) does not include a heritability estimate of income. The reason we included this reference is because it discusses the similarities and differences between heritability estimates from family studies (e.g. the classic twin design) and heritability estimates based on genomic data (e.g. GREML). Heritability estimates from genomic data are typically lower than in family studies for several reasons, including the fact that SNP data only cover a part of the total genetic variation between individuals (i.e. excluding structural and rare genetic variants).

2) Although time-specific intercepts were employed, how authors controlled the effects of inflation to ensure comparability between longitudinal data and cross-sectional survey data?

The cohort-specific GWAS results were meta-analysis using sample-size weighting (SI p. 8), i.e. we meta-analyzed the Z-statistics of each SNP rather than effect size estimates (betas).

At the cohort level, our preregistered analysis plan (<https://osf.io/7z45j>, p. 10-11) contained separate instructions for cross-section and panel data analyses. In both cases, the effect of inflation was controlled by including dummy variables for the year of observation. Panel data (i.e. multiple observations of income per individual) were averaged over all observation periods after residualizing for year of observation dummy variables.

3) According to the current results from Supplementary Table 22, I can't deduce how these estimates were developed.

a) "These results imply that only 24.0% or 25.7% of the INC factor PGI's predictive power was due to direct genetic effects, which was very close to the result for the EA PGI estimated elsewhere (25.5%).³⁸"

According to Young et al. (2022), the fraction of variance explained by the direct effect is the square of the ratio of direct-to-population effect. Thus, the results from Supplementary Table 22 implies the INC factor PGI's predictive power for household and occupational income are $(0.09467/0.19342)^2 = 0.240$ and $(0.11127/0.21962)^2 = 0.257$, respectively.

4) Please provide the proportion of original parental genotypes, imputed parental genotypes with observed data, and imputed parental genotypes with the frequency (f) of allele 1.

In UKB, we found 19,353 pairs of full siblings and 5,342 parent-offspring pairs. The average number of shared non-missing SNPs that passed INFO and MAF filtering among these pairs is 544,256 ($STD=2,603$), which accounts for 43.75% of the HapMap3 SNPs (Supplementary Information section 5.2).

In STR, we did not use any imputed parental genotypes since we don't have income data for parents in the genotyped sample. The proportion of both genotyped and imputed parents is therefore zero in this sample.

5) Page 21, there is a clerical error in the Neff of the GWAS in USA.

Thank you, we fixed this ($N_{eff} = 30,855$).

6) The total results section lacks a clear structure. For instance, in the polygenic prediction, the author has actually covered a substantial amount of content, including estimates of direct effect proportions and PheWAS. These results address different scientific questions, but the author just combined them in a crude way without proper organization. Then, they moved on to genetic correlation, which is more closely related with the previous section (Comparison with educational attainment), both in terms of research methods and content.

We improved the organization and readability of the paper. The results section is now structured as follows:

1. Multivariate GWAS of income
 - GWAS of four different measures of income
 - The Income Factor
 - Identification of genetic loci
 - Effect sizes
 - Cross-sex and cross-country heterogeneity
2. Comparison with educational attainment
 - Genetic correlation with educational attainment
 - GWAS-by-Subtraction
3. Polygenic prediction

4. Direct vs. indirect genetic effects
5. Income and health
 - Genetic correlations with psychiatric and health traits
 - Phenome-wide association study (PheWAS) on electronic health records
6. Biological annotation

This structure provides a logic flow and clearly separates different analyses that address distinct scientific questions.

7) Figures provided by the author were incomplete.

We apologize. This probably happened during the PDF assembly stage in the submission system. We double-checked that all Figures are included in the resubmission package.

8) Authors needs to add explanation why the genetic correlations estimated were greater than 1. Did it imply data quality is actually very poor?

The possibility of obtaining genetic correlation estimates of >1 with LDSC is a well-known property of the method. For example, the original paper (Bulik-Sullivan 2014, doi:10.1038/ng.3406) contained various r_g estimates >1 in Supplementary Table 4 (e.g., 1.467 for fasting insulin and HOMA-IR).

The support page for the LDSC software (<https://github.com/bulik/ldsc/issues/78>) explains this phenomenon: “The estimate of r_g is unbounded and consists of the true signal plus random error. When sample overlap is high, this error becomes greater. Similarly, when two highly similar traits are analyzed, there is an increased chance that the estimated r_g will exceed 1.” Thus, this is not a sign of low data quality.

9) Personally, I don't think the title is accurate. The incremental R2 didn't reflect the relationship between polygenic score and income. Authors needs to provide a detailed distribution of polygenic scores, e.g correlations with income or in the form of percentiles, to better capture the “gradient. It's a necessary step before the phenome-wide association study of the INC factor PGI on socioeconomic health.

We added Extended Fig 3.b, which shows average levels of individual/occupational/household income per PGI quintile in STR, along with 95% confidence intervals. The analyses contain $N = 28,359 / 21,990 / 17,418$ observations respectively. Outcome variables were residualised on sex and the first 20 principal components. The residuals have a mean of zero and a standard deviation of one. Predictive accuracy is highest for individual income, which is also the best measure of income available (derived from Swedish registry data). Prediction accuracy is lowest for household income, with a difference of ~ 0.15 standard deviations between the lowest and highest quintile of the PGI distribution.

We added the following text to the Polygenic Prediction section of the main manuscript:

“Extended Fig 3b shows average income levels per PGI quintile in the STR sample. The expected income of individuals increases monotonically for higher PGI quintiles. Predictive accuracy is highest for individual income, the most accurate measure of income (derived from Swedish registry data). The difference in average income for individuals in the lowest and highest quintile of the PGI distribution is ~0.2 standard deviations.”

Furthermore, the following scatter plot is included in the FAQ on p. 13, with a discussion of the results on p. 12:

Note: The figure presents a scatter plot for the Health and Retirement Study ($N=6,171$). The x-axis shows standardized values of the Income Factor PGI, constructed from a GWAS meta-analysis that excluded the Health and Retirement Study. The y-axis shows log self-reported income, which was constructed as follows: within each wave, the log income was regressed on demographic variables (sex, age, age², age³, and the interactions between sex and the age terms) and genetic control variables (top 20 genetic principal components and genotyping batches). Then, the mean of the residuals from the regressions was obtained for each individual, which was then standardized. The dotted line is a regression line with slope 0.172 ($p = 5.3 \times 10^{-42}$).

10) this research was pre-registered on August 30 2018. And the study is generally consistent with the protocol. However, the INC factor was not included in the protocol. Moreover, the correlation between the constructed INC factor and the actual income was not explained in the manuscript.

Correct, we pre-registered our analysis plan on 30 Aug 2018. At that time, the genomic structural equation modelling (GSEM) method was not published yet. The GSEM method was published almost a year later, in April 2019 (see <https://www.nature.com/articles/s41562-019-0566-x>). This and the ex-ante unknown genetic correlations between the various measures of income included in our study were the main reasons we did not pre-commit to run GSEM analyses in the analysis plan.

Fig 1b includes genetic correlation estimates between the INC factor and all other income measures. Specifically, the INC factor has a genetic correlation with individual income indistinguishable from 1. We now mention this finding in the main manuscript.

Reviewer #2:

Remarks to the Author:

The authors conducted a GWAS of income on individuals of all European ancestry. Although the study has strength of big sample size, I have several major concerns about the study design and GWAS methodology. Below are my specific comments:

The biggest issue of this study is the bias coming from phenotype (i.e., income) and population ancestry (i.e., European). It is well-known that race/ethnicity is strongly related to economic status (e.g., income). However, this study only investigated the European population, which cannot truly reflect the GWAS signal for income due to the lack of other races/ethnicities.

Although the same size is huge, most samples are from UK Biobank. This study is a metaanalysis (i.e., not single cohort GWAS), I think it will be nice to have individual GWAS of other ancestries, then meta-analysis with European ancestry.

This will benefit from studying population-specific GWAS signals and comparing GWAS signals across different population ancestry. This is especially important for the income phenotype.

We appreciate the reviewer's suggestion and agree that race/ethnicity is strongly related to income in many countries. However, one of our study's main findings is the considerable heterogeneity in the genetic architecture of income across the included European populations. Specifically, we examined cross-cohort genetic correlations and found that the inverse-variance weighted mean genetic correlations across pairs of cohorts were 0.45 (s.e. = 0.22) for individual income, 0.52 (s.e. = 0.13) for household income, and 0.90 (s.e. = 0.24) for occupational income (Supplementary Tables 28a-c).

Furthermore, we meta-analyzed cohorts from the same country with the same income measure available and estimated the genetic correlations across these countries. We found that occupational income in Norway displayed lower genetic correlations with occupational or

household income in other countries, ranging from only 0.43 (s.e. = 0.23) to 0.82 (s.e. = 0.10). Similarly, occupational income's genetic correlation with educational attainment (EA) was also lower in Norway ($r_g = 0.69$, s.e. = 0.08) compared to the other countries.

This series of analyses evinces the heterogeneity of GWAS on income across European samples and casts doubt on the idea that a "true," universal genetic architecture of income exists. By including samples with different genetic ancestries or other cultures where this signal may be very different, these sample-specific associations would be lost: While meta-analysis increases statistical power and yields more precise estimates of the average effect size, it also tends to mask non-random heterogeneity in effect size estimates across samples.

Inspired by the reviewer's comment, we clarified the discussion section of the manuscript as follows:

"It is important to point out that the results of our study reflect the specific social realities of the analysed samples and are not universal or unchangeable. This is exemplified by the substantial heterogeneity in the genetic architecture of income that we found across our cohorts of European descent, as well as the non-perfect genetic correlation between sexes. This heterogeneity is consistent with previous findings where the polygenic signal for other measures of SES (such as educational attainment) varies by culture (Rimfeld et al. 2018, <https://www.nature.com/articles/s41562-018-0332-5>) and by country (Tropf et al. 2017, <https://www.nature.com/articles/s41562-017-0195-1>). This genetic heterogeneity is indicative of phenotypic heterogeneity between cultures, where the heritable traits linked to income may not be universal but rather vary and reflect the differences between societies in which heritable traits are facilitative of income differences.

We emphasise that our results are limited to individuals whose genotypes are genetically most similar to the EUR panel of the 1000 Genomes reference panel compared to people sampled in other parts of the world. Our results have limited generalizability and do not warrant meaningful comparisons across different groups or predictions of income for specific individuals (FAQ). To increase the representation of individuals from diverse backgrounds, cohort and longitudinal studies should seek to sample more diverse and representative samples of the global population."

However, the latter and the inclusion of non-European samples are beyond the scope and possibilities of the current project.

I am not clear why age3 needs to be adjusted in the model for revisualization. I have seen age and age2, but it is very rare to adjust age3 unless it is well-justified.

We chose to include *Age*³ as a control variable because the relationships between age, income, and genotypes are likely to be highly complex. For example, individual income varies over time not only as a function of labour market experience, performance, and seniority, but also due to business cycle dynamics and long-term economic growth. Furthermore, age may also be related to the observed genotypes in a sample due to survival effects: Most cohorts included in

our study comprise a substantial proportion of elderly individuals. The probability of having them included in a study depends on whether they survived long enough and are healthy enough to participate, which in turn can depend on their (health- and longevity-related) genotypes. It is ultimately an empirical question if Age and Age^2 are sufficient as control variables, or whether Age^3 needs to be included as well.

In any case, including Age^3 in addition to Age and Age^2 as control variables is a conservative approach that protects against potential biases. In particular, if Age^3 is relevant but not included, the GWAS results would be biased. If Age^3 is relevant and included, no bias is expected, but the variance of the estimates increases, which leads to higher p -values. Finally, if Age^3 is irrelevant but included, the estimates remain unaffected.

To see why, consider the standard ordinary least squares regression (OLS) model **without** the cubic age term

$$Y = \beta_0 + \beta_1 X_1 + \beta_2 G + \epsilon,$$

where Y is the phenotype; X_1 is the covariates matrix; G is the genotype; and ϵ is the error term. The variance of β_2 estimated with OLS is given by

$$Var(\hat{\beta}_2) = \frac{\sigma^2}{\sum (G - \underline{G})^2},$$

where σ^2 is the variance of error term, G is the genotype, and \underline{G} is its mean.

By including the cubic age term into the model, we obtain the following model:

$$Y = \beta_0 + \beta_1 X_1 + \beta_2 G + \beta_3 Age^3 + \epsilon.$$

Under the classical linear regression assumptions, OLS estimators are unbiased, meaning that on average, across many samples, they correctly estimate the true population parameters. This property holds even when irrelevant variables are included because the expected estimate for β_3 is 0. Including Age^3 changes the variance of the estimated β_2 to

$$Var(\hat{\beta}_2) = \frac{\sigma^2}{\sum (G - \underline{G})^2 (1 - R_{G, Age^3}^2)},$$

where R_{G, Age^3}^2 is the coefficient of determination between the genotype and Age^3 .

If G and Age^3 are uncorrelated, then both the estimate of β_2 and its variance are the same as previously. Thus, including Age^3 as an irrelevant control variable does not change the results.

If, however, G and Age^3 as well as Y and Age^3 are correlated, the omission of Age^3 would lead to omitted variable bias of estimates of β_2 . Furthermore, the variance of the β_2 estimates would become larger because the denominator multiplies with the result of one subtracting the proportion term, which is a decimal smaller than one (Greene, 2003; Wooldridge, 2009). A

larger variance of β_2 implies lower t -statistics, which in turn imply that the chance of finding p -values $< 5 \times 10^{-8}$ decreases. Thus, including Age^3 as a control variable is a conservative approach.

After GWAS meta-analysis, it is unclear whether the signals are LD-independent or not. It seems like the authors did not perform a clumping analysis.

We can confirm that clumping was used to identify independent loci. Specifically, LD-based clumping was performed using FUMA. The details of how FUMA is used to identify independent loci can now be found in the method section titled "GWAS meta-analysis." The additional text now included has been copied below for convenience.

"Independent loci were identified using FUMA (<https://www.nature.com/articles/s41467-017-01261-5>). First, independent significant SNPs were defined using a cut-off of $P < 5 \times 10^{-8}$ and as independent from any other SNP ($r^2 < 0.6$) within a 1-mb window. Next, lead SNPs are identified as significant SNPs independent from each other at $r^2 < 0.1$. Finally, independent genomic loci are formed from all independent signals that are in physical proximity to each other by merging independent significant SNPs closer than 250kb into a single locus using the 1000 genomes EUR reference panel to ensure the accuracy of the loci borders were not influenced by missing data in our GWAS. As such, the distance between two loci defined by FUMA is between the SNPs in LD with the independent significant SNPs rather than between the independent significant SNPs themselves."

In addition, a conditional analysis needs to be performed to identify secondary loci that are conditioned on the primary loci, such as using COJO.

We ran Conditional and Joint Association Analysis (COJO) using the Genome-wide Complex Trait Analysis (GCTA) software to refine our understanding of the genetic architecture underlying the trait of interest (J. Yang et al., 2012). The analysis was performed with a window size of 100,000 base pairs (bp), conditioned on 207 primary lead SNPs from 162 loci, previously identified as significantly associated with the Income Factor. Our COJO analysis revealed 57 secondary lead SNPs that surpassed the Bonferroni corrected threshold for statistical significance ($p \leq 5 \times 10^{-8}$), conditioning on the primary lead SNPs. Notably, 55 of these secondary lead SNPs were located within the original primary genomic loci, underscoring their potential role in the same genetic regions initially implicated in the association with the Income Factor. The remaining two secondary lead SNPs were identified as novel loci. See details of these secondary lead SNPs in Supplementary Table 30 and section 2.6 of the Supplementary Information.

We added the following section to the main manuscript (section *Identification of Genomic Loci*):

"Furthermore, we conducted Conditional and Joint Association Analysis (COJO) on 207 lead SNPs associated with the Income Factor (J. Yang et al., 2012), revealing 57 secondary lead SNPs ($p \leq 5 \times 10^{-8}$). 55 of these secondary lead SNPs were located within the original primary genomic loci (Supplementary Table 30, Supplementary Information 2.6)."

The fine-mapping analysis is also missing from this study, but this is standard for GWAS and has to be done.

We created Locus Zoom plots for all lead SNPs and made them available at <https://beta.dpid.org/149> (folder Locus Zoom plots).

Furthermore, we applied CARMA, a Bayesian fine-mapping method, to identify the most likely causal SNP within a locus. CARMA is reported to outperform existing methods by achieving lower false discovery rates (FDR) and greater statistical power, making it especially useful for GWAS meta-analyses involving multiple cohorts (Z. Yang et al., 2023). Specifically, we used CARMA to deduce each SNP's posterior inclusion probability (PIP) across 162 genomic loci associated with the Income Factor.

The reference panel for calculating linkage disequilibrium (LD) was the Haplotype Reference Consortium (HRC) (<https://ega-archive.org/datasets/EGAD00001002729>). We performed genotype and sample quality control (QC) by keeping only bi-allelic SNPs with a minor allele frequency above 1% (with option `--maf 0.01`), filtering out related individuals (with option `--rel-cutoff 0.025`) and samples with any missing genotypes (with option `--geno 0`). These filtering steps reduce the sample size from 22,691 to 17,774 unrelated individuals, resulting in a high-quality reference panel suitable for subsequent fine-mapping analysis.

Likely causal candidates SNPs with the highest PIP in each locus are reported in Supplementary Table 31. Notably, six SNPs were found to have a PIP greater than 0.95, suggesting a high probability of being causal for the Income Factor. These SNPs are rs17571877 (Locus 4, PIP=1.00), rs13022707 (Locus 18, PIP=1.00), rs9821311 (Locus 32, PIP=1.00), rs13107325 (Locus 41, PIP=0.99), rs77543296 (Locus 45, PIP=1.00), and rs59378495 (Locus 69, PIP=0.99). Among these variants, rs13107325 is a (benign) missense variant of SLC39A8 that was previously reported in more than 50 publications (<https://www.ncbi.nlm.nih.gov/snp/rs13107325#publications>). The other five causal SNP candidates are intronic variants with unknown clinical significance.

The results of functional analysis from this study are only confirmatory to the previous findings (references 21 and 24), and not novel. In addition to central nervous system, any new biological function/tissue/system was identified?

The referee is correct that our original analysis based on LDSC partitioned heritability analyses did not yield any novel insight. Given that our phenotype is a social construct rather than a disease or a physiological characteristic, we consciously decided to keep the biological annotation of our GWAS results to a minimum in the original draft of the paper.

However, we now performed additional biological annotations to address this question.

Because Lee et al. (2018), Hill et al. (2019), and our study used different methods to define candidate genes associated with their target traits, we re-ran gene-based analyses in MAGMA

using summary statistics from all three studies. (Note that 23andMe data are excluded in the public release of Lee et al. summary statistics). Then, we extracted genes significantly associated with the three target traits at $P.adjust < 0.05$ (Bonferroni correction for each study). The overlap among these genes is shown below. Although EA3 (Lee et al. 2018) has much greater statistical power due to its larger sample size, we identified 98 genes for our Income Factor that were not previously discovered to be associated with income or education.

Using FUMA GENE2FUNC, we further examined the biological processes of these genes. The overlap among biological processes detected for each trait at FDR < 0.05 is shown below. We found three processes that were only discovered for our Income Factor:

GOBP_NEURON_MIGRATION (fdr = 0.012),
GOBP_ENDOCHONDRAL_BONE_MORPHOGENESIS (fdr = 0.036), and
GOBP_REGULATION_OF_AXONOGENESIS (fdr = 0.047).

There are also three processes that were missed in earlier income GWAS (Hill et al. 2019), but were detected in EA3 (Lee et al. 2018):

GOBP_CELL_JUNCTION_ORGANIZATION (FDR = 9.8×10^{-5}),
GOBP_DENDRITE_DEVELOPMENT (FDR = 3.66×10^{-3}), and
GOBP_TELENCEPHALON_DEVELOPMENT (FDR = 0.047).

We added the following paragraph to the main manuscript in response to the reviewer's comment:

“Next, we compared the genes identified with MAGMA for the Income Factor with those identified previously for EA and household income. We find that of the 368 genes associated with the Income Factor, 98 were not discovered for educational attainment or household income yet (Extended Fig. 5b (a) & Supplementary Tables 32-34). We further examined the biological processes of genes associated with the Income Factor, EA, and household income with FUMA GENE2FUNC. Using a test of overrepresentation, we find three biological processes at FDR < 0.05 that are unique to the Income Factor: neuronal migration (FDR = 0.012), bone formation in early development (FDR = 0.036), and the formation of axons (FDR = 0.047). The overlap among biological processes detected for each trait at FDR < 0.05 is shown in Extended Fig 5b (b) (Supplementary Tables 35-37).”

In the Supplementary Information, we have added the following text:

“4.1. Biological annotation

To examine if the Income Factor was capturing the same underlying biology as household income and educational attainment, we used MAGMA and a test of overrepresentation performed using the GENE2FUNC process in FUMA (version 1.5.2). First, gene-based statistics were derived for the INC factor and educational attainment (Lee et al. 2018) using MAGMA. For household income, the MAGMA gene-based statistics were taken from Supplementary Table 18 in Hill et al. (2019). Next, genes that passed a Bonferroni correction were retained and compared across the Income Factor, educational attainment (Lee et al. 2018), and household income (Hill et al. 2019). This comparison can be seen in the Venn diagram in Extended Fig 5b (a).

Second, using the GENE2FUNC in FUMA, we performed a hypergeometric test to determine if the genes identified using MAGMA were overrepresented in biological pathways using MsigDB. Gene sets that attained statistical significance (FDR <0.05) in the Income Factor, educational attainment, and household income were retained and compared against each other (Extended Fig 5b (b)).”

The extended figure 5 was cut in half, and not showing these results.

We apologize for the inconvenience. This might have happened during the PDF assembly in the submission system. We double-checked that all Figures are included in the submission package.

In addition, all analyses were based in silico, it will be much stronger to have more mechanistic data (e.g., in vitro, in vivo) to validate the function of identified loci.

Given that our phenotype is a social construct rather than a disease or a physiological characteristic, we consciously decided to keep the biological annotation of our GWAS results to a necessary minimum.

The abbreviation “INC” used in this manuscript is not very helpful because it does not shorten the whole word; instead, it just represents income. So why not just use “income”, which is clearer?

We changed the wording from “INC factor” to “Income Factor” throughout and from “NonEA-INC” to “NonEA-Income.”

Reviewer #3:

Remarks to the Author:

I trust this letter finds you well. I am writing to convey the positive outcome of the review process for your manuscript. The quality of your research, the robustness of your methodology, and the clarity of presentation have all contributed to this positive recommendation.

The study is well-organized, the literature review is comprehensive, and the conclusions drawn are supported by the presented data. While the manuscript is well-documented as it stands, I would like to highlight some minor suggestions for improvement that you may consider addressing.

(1) In the first paragraph of results section, four distinct measures of income were delineated. Subsequently, independent GWAS were performed on each of these measures, leading to the identification of 86 non-overlapping loci. However, the biological functions of these loci remain unclear, and it is currently unknown how many loci are shared among the four income measures.

We added the following Venn diagram of the 86 non-overlapping loci across the four distinct measures of income as Extended Fig 5c:

Next, we extracted high-confidence genes mapped to those 86 loci. Gene-based statistics were derived using MAGMA for genes whose physical boundaries overlapped with these genome-wide significant loci. A Venn diagram of the derived genes across the four income measures is shown below and in Extended Fig. 5c (b).

Next, we took 63 genes unique to occupational income, 24 unique household income genes, and 55 genes shared between occupational and household income into FUMA GENE2FUNC for gene-set enrichment analysis to evaluate their differences and similarities in biological annotations. We found that shared genes are down-regulated in blood vessels, while genes unique to household income are up-regulated in the brain and nerves. Genes unique to household or occupational income were not found to be enriched for any biological processes, cellular components, or molecular function gene sets. In contrast, shared genes are enriched to regulate the modification of synaptic structure and synapse organization gene sets. These results are shown in the table below.

Gene Set Category	Occupational Unique Genes	Shared Genes	Household Unique Genes
GTEX Expression	0	Down: Blood Vessel	Up: Brain & Nerve
Biological processes	0	2 REGULATION_OF_MODIFICATION_OF_SYNAPTIC_STRUCTURE & SYNAPSE_ORGANIZATION	0
Cellular components	0	0	0
Molecular functions	0	0	0
GWAS Catalog 	40 Top 3 Traits: Intelligence, Subcortical Volume, Brain morphology	66 Top 3 Traits: Extremely High Intelligence, Brain Morphology, Sleep Duration	16 Top 3 Traits: Household Income, Intelligence, Cognitive Ability

Note that the differences we identified across the four income measures could be driven by differences in statistical power and measurement accuracy rather than by different biological processes.

This figure shows the p-value for trait-associated loci across traits. The Y-axis represents the minimum p -value of association of any SNP within a locus from a certain GWAS. The X-axis represents the trait-associated loci from of which traits. Taking the third column (occupational) as an example: Cells in the top row are colored in dark red because these loci are occupational income loci. The second row represents the strongest SNP association of these occupational

loci in household GWAS. Some are marginally significant in household GWAS (coloured in yellow), some passed the suggestive significance level (coloured in orange), and some are GW-significant (coloured in dark brown).

Looking at the plot vertically, we find that for any occupational income locus, there is at least one statistical evidence that SNPs in this region is associated with another income index. Hence, each locus in the plot is at least associated with 2 indices at $p < 0.05$, suggesting the signal discovered from four different income measures is quite homogeneous.

We have now amended the manuscript to include this additional information regarding how the significant loci and the genes within these loci overlap between the four income traits. The additional text is copied below.

“Across the four GWAS on different income measures, we identified 86 non-overlapping loci in the genome (see Supplementary Information section 2.6 for the definition of loci and lead SNPs, and Extended Fig 5c (a) for the distribution of associated loci across the four income traits). Table 1 summarises the results. Occupational and household income showed the most genetic associations (59 and 41 loci, respectively), as expected based on sample sizes and SNP-based heritability estimates based on linkage disequilibrium score regression (LDSC) ($h^2 = 0.08$ (s.e. = 0.003) and 0.06 (s.e. = 0.003), respectively). Gene-based analysis was performed on the genes that overlapped with each loci using MAGMA, where 102 attained genome-wide significance, with 63 being unique to occupational income, 24 unique to household income, and 55 shared between the two. No other genes attained statistical significance (Extended Fig. 5c (b)).”

(2) In the analysis of "Cross-sex and cross-country heterogeneity," were the cohorts included in the meta-analysis exclusively of European ancestry?

Yes. All individuals and samples included in this study are of European descent.

(3) Additional evidence is required to substantiate the functional role of rs34177108, which was identified through the NonEA-INC GWAS.

To examine the functionality of the NonEA Income locus, we extracted the three lead SNPs from this locus. We compared the association test statistics across educational attainment, the Income Factor, a GWAS on red vs. non-red hair, and a GWAS on blond vs. non-blond hair. We

found the A allele of rs34177108 is associated with a decrease in NonEA-Income ($\beta = -0.013$, $SE = 0.002$) as well as a negative effect on income that is not genome-wide significant ($\beta = -0.010$, $SE = 0.002$, $P = 1.44 \times 10^{-6}$). There was no evidence that this A allele had an effect on educational attainment ($\beta = 0.001$, $SE = 0.002$, $P = 0.51$). However, this allele was associated with a greater instance of red ($\beta = 104$, $SE = 0.001$, $P = 0$) and blond hair ($\beta = 0.023$, $SE = 0.001$, $P = 5.68 \times 10^{-157}$) than all other hair colors. This pattern was directionally consistent across the three lead SNPs, and rs1805007 retained statistical significance for the income factor (Supplementary Table 38).

This pattern of associations across non-EA Income, the Income Factor, educational attainment, and red and blond hair indicates that the red and blond hair genotypes at this locus have no effect on educational attainment. However, they are related to a lower income level.

We next examined the relationship between the broader polygenic signal for red and blond hair with Non-EA Income, educational attainment, and the Income Factor. We used publicly available GWAS summary data on hair color (Supplementary Table 39) from the IEU GWAS database (<https://gwas.mrcieu.ac.uk/>). These include three case-control phenotypes (Red vs Non-Red, Blonde vs Non-Blond, and Dark Brown vs Non-Dark-Brown) from Neale et al. (<https://www.nealelab.is/uk-biobank>), two case-control phenotypes (Blond vs Non-Blond labeled as HC_Blond and Dark Brown vs Non-Dark-Brown labeled as HC_DarkBrown) from Loh et al. (<https://www.ncbi.nlm.nih.gov/pmc/articles/PMC6309610/>), and one integer phenotypes from Loh et al. (labeled HC and recorded from light hair color (higher numbers) to dark hair color (lower numbers)).

Using linkage disequilibrium score regression, we found little evidence of a sufficient polygenic signal to examine red hair ($h^2 = 0.018$, $SE = 0.013$, $Z = 1.31$), as evidenced by a heritability Z score of less than 4. The heritability of each other category of hair color could be measured with sufficient accuracy to perform genetic correlations (Supplementary Table 40).

Significant genetic correlations were present between the Income Factor with blond ($r_g = -0.06$, $SE = 0.02$, $P = 0.015$), dark brown ($r_g = 0.012$, $SE = 0.025$, $P = 3.35 \times 10^{-6}$), HC ($r_g = -0.06$, $SE = 0.02$, $P = 0.006$), HC_blond ($r_g = -0.053$, $SE = 0.021$, $P = 0.012$), and HC_DarkBrown ($r_g = 0.120$, $SE = 0.024$, $P = 4.12 \times 10^{-7}$). The direction of association across each of these traits is consistent with the idea that the genetic antecedents of darker hair are also associated with higher income.

This genetic relationship between darker hair and a greater level of income was not significant for Non-EA Inc as evidenced by the genetic correlations with blond ($r_g = -0.077$, $SE = 0.047$, $P = 0.102$), darkbrown ($r_g = 0.071$, $SE = 0.049$, $P = 0.152$), HC ($r_g = -0.051$, $SE = 0.038$, $P = 0.173$), HC_blond ($r_g = -0.050$, $SE = 0.043$, $P = 0.248$), and HC_DarkBrown ($r_g = 0.058$, $SE = 0.045$, $P = 0.201$). However, the direction of these genetic correlations was consistent with those of the Income Factor.

Genetic correlations with red hair are also presented in Supplementary Table 41 for completeness. Whilst these are directionally consistent with the findings presented above (i.e. lighter hair is genetically related to a lower level of income), no strong conclusions should be drawn from these data due to the poor accuracy of the heritability that could be estimated.

As a next step, we investigated relationships between hair color and income using phenotyping data from UK Biobank (data-field 1747). Regression analyses showed that individuals with black hair would be more likely to have lower income (relative to blond hair), while those with dark brown hair are more likely to have higher income (see Response Letter Table 1 below).

Response Letter Table 1

HI ~ Hair Colour	Estimate	Std.	Error	t value	Pr(> t)
(Intercept Blond)	2.595	0.006	450.470	$<2 \times 10^{-16}$	***
Red	0.004	0.011	0.391	0.695	
LightBrown	0.009	0.007	1.323	0.186	
DarkBrown	0.066	0.007	10.024	$<2 \times 10^{-16}$	***
Black	-0.138	0.011	-12.546	$<2 \times 10^{-16}$	***

Note: Only participants of European descent were included.

Finally, we included educational attainment in these regressions, allowing for an interaction between hair color and education on income. Our results show that the effect of education on income is greatest for those with black hair. In contrast, those with dark brown benefitted the least from a university-level degree, leading to a greater similarity of income in those attaining a university-level degree compared to those who did not attain a university degree (Response Letter Figure 1).

Overall, this highly complex -- and partially contradictory -- set of results does not point to a clear story about discrimination. A further investigation of this finding is beyond the scope of this paper. Therefore, we simply deleted “discrimination” as a potential explanation of this finding. The GWAS-by-Subtraction section of the main manuscript now reads as follows:

“We employed the GWAS-by-subtraction approach using Genomic SEM33 to identify this residual genetic signal (referred to as ‘NonEA-Income’). We identified one locus of genome-wide significance for NonEA-Income, marked by the lead SNP rs34177108 on chromosome 16 (Extended Data Fig 2c). This locus was previously found to be associated with vitamin D levels,

cancer, as well as hair and skin-related traits such as colour, sun exposure, possibly picking up on uncontrolled population stratification (Supplementary Tables 38-41).”

Response Letter Table 2.

HI ~ HC + EA + HCxEA	Estimate	Std.	Error	t value	Pr(> t)
(Intercept Blond)	2.313	0.007	348.888	$<2 \times 10^{-16}$	***
Red	-0.007	0.013	-0.547	0.585	
LightBrown	0.008	0.007	1.083	0.279	
DarkBrown	0.047	0.008	6.200	5.66×10^{-10}	***
Black	-0.120	0.012	-9.696	$<2 \times 10^{-16}$	***
EA	0.872	0.012	75.369	$<2 \times 10^{-16}$	***
Red:EA	-0.027	0.021	-1.261	0.207	
LightBrown:EA	-0.015	0.013	-1.182	0.237	
DarkBrown:EA	-0.035	0.013	-2.638	0.008	**
Black:EA	0.061	0.023	2.688	0.007	**

Table 2. The results of multiple regression examining hair color (HC) and educational attainment (EA) on household income (HI). Education is coded as 1 for those with a university-level degree and 0 for those without. Participants were included if they were of European descent.

Response Letter Figure 1.

Figure 1. This figure shows how income differences between different hair color groupings are smaller in those with a higher educational attainment than those with a lower level of education. First, the average income level was derived for the two education groupings. Second, the average income level was derived for each hair color group separately in the two education groupings. Third, the average income level of each education grouping was subtracted from the income level within each hair color grouping. Each bar shows the average income level of each hair color group following the subtraction of the average income level within the two education groupings.

Reviewer #1 (Remarks to the Author):

The authors have addressed the majority of previous comments satisfactorily. The improvements in methodology, data presentation, and overall clarity are commendable. I have a minor concern regarding the direct and indirect genetic effects. Although this issue occupies only a small portion of the results, it is worth discussing in the context of the overall conclusions. According to the data provided by the authors, only about 20% of the study subjects have complete nuclear family data, while the rest were imputed. In snipar, not all situations can accurately infer parental genotypes (IBD0). In other instances (when siblings share one allele IBD or both alleles IBD), parental alleles are unobserved and imputed with the frequency of allele 1. Therefore, I would recommend that the authors provide information on the quality control of the imputation, such as genotyping error rate, proportion of IBD0, or perform a sensitivity analysis on the nuclear families with complete information.

The quality control procedure we used for snipar is described in Section 5.2 in the Supplementary Text. We added the information on the genotyping error rate and the proportion of IBD0 as follows:

[...] The average observed proportion of IBD0 pairs was 22.3%, which closely matched the expected proportion of 22.5% given that the sibling pairs were the majority. The estimated genotyping error was also sufficiently low, with an average of 0.07%.

Overall, I am satisfied with the revisions made. Thank you again for the opportunity to review this manuscript. It has been a valuable learning experience for me.

Thank you!

Reviewer #2 (Remarks to the Author):

Overall, the authors addressed most of my comments. However, several issues still need to be addressed:

1. In the current era of GWAS, it is not always bigger and better. After many years of GWAS, it is more important to study non-European populations than European populations. The authors made a claim that it is out of the scope to study non-Europeans in the current study. This is not true. UKB has many non-European subjects with decent sample size. I still suggest the authors include those subjects for the population specific analysis.

The UK Biobank has a self-reported ethnic background data field (21000), from which we extracted samples that were self-reported as African, Caribbean, or Indian. According to these self-reports, the three largest groups of non-Europeans in the UKB are Africans ($N = 3,457$), Caribbean ($N = 4,600$), and Indian ($N = 6,104$). Given the extremely small effect sizes of individual

SNP for income ($R^2 < 0.01\%$), these sample sizes are too small to conduct a statistically well-powered GWAS analysis.

Although we could not conduct discovery GWAS analyses on non-European samples, the revised version of the paper now includes polygenic prediction analyses that we carried out in these non-European samples. In addition to using self-reported ethnic backgrounds, we also conducted these analyses using genetic ancestry groups derived from the SNP data (Africans $N = 9,494$, East-Asians $N = 2,216$, and South-Asians $N = 11,413$).

We found the following results:

Self-reported ethnic backgrounds. For household income, the predictive accuracy of the PGI for African, Caribbean, and Indian ancestry groups were $\Delta R^2 = 0.40\%$ (95% CI: 0.00% - 0.73%), $\Delta R^2 = 0.64\%$ (95% CI: 0.08% - 1.06%), and $\Delta R^2 = 1.62\%$ (95% CI: 0.94% - 2.33%), respectively. For occupational income, the predictive accuracy of the corresponding PGI were $\Delta R^2 = 0.85\%$ (95% CI: 0.10% - 1.44%), $\Delta R^2 = 0.16\%$ (95% CI: 0.00% - 0.32%), and $\Delta R^2 = 1.86\%$ (95% CI: 1.13% - 2.57%), for African, Caribbean, and Indian ancestry groups, respectively.

Genetic ancestry. For household income, the predictive accuracy of the PGI for African, East Asian, and South Asian ancestry groups were $\Delta R^2 = 0.38\%$ (95% CI: 0.07% - 0.61%), $\Delta R^2 = 0.09\%$ (95% CI: 0.00% - 0.17%), and $\Delta R^2 = 1.58\%$ (95% CI: 1.05% - 2.06%), respectively. For occupational income, the predictive accuracy of the corresponding PGI were $\Delta R^2 = 0.45\%$ (95% CI: 0.10% - 0.71%) for African group, $\Delta R^2 = 0.16\%$ (95% CI: 0.00% - 0.32%) for East Asian group, and $\Delta R^2 = 1.34\%$ (95% CI: 0.85% - 1.84%) for South Asian group.

Thus, as expected based on earlier studies, polygenic indices derived from GWAS results on European samples have substantially reduced (but non-zero) predictive accuracy in non-European populations.

2. Some careless mistakes still present. For example, the authors made a claim that they have changed the wording from "INC factor" to "Income Factor" throughout the manuscript. However, Figures 1 and 2, and many other figures still show "INC factor". This shows that the authors did not pay attention to details.

Thank you for spotting that! We fixed this and carefully proofread all materials again.

Reviewer #3 (Remarks to the Author):

I am pleased to inform you that I find it to be of high quality with no further revisions needed.

Thank you!

Reviewer #4 (Remarks to the Author):

The authors conducted a GWAS on income in the individuals of European descent, identifying 162 genomic loci associated with a common genetic factor underlying various income measures. Despite the small effect sizes, these findings offer valuable insights into the complex relationship between genetics, income, and health outcomes. The study's polygenic index captures 1-4% of income variance, highlighting the modest but noteworthy genetic influence on income. Several post-GWAS analyses, including PheWAS, further elucidate the health implications, demonstrating reduced risks for several diseases among individuals with a higher polygenic index for income. While the study's methodological comprehensive approach are commendable, I have some concerns and suggestions.

Major concern:

Considering the high genetic correlation with previous household income GWAS or educational attainment GWAS, I have concerns about the incremental value of the current study. The study identifies 162 genomic loci associated with income, yet these loci have miniscule effect sizes and explain only few portions of the income variance.

Although the lack of novel signals compared to previous GWASs might seem like a limitation, I believe these findings still contribute to the field. It would be beneficial to emphasize the importance of these findings in the context of existing sociogenomic literature and discuss the implications of these "modest" findings, like how they advance our understanding of the genetic architecture of SES. How do these results compare with other socioeconomic traits?

We appreciate the reviewer's comments and suggestions. We have amended the discussion to highlight our finding that the genetic architecture of income is not just a simple

subset of educational attainment but is likely to contain effects that are unique to it. The amended text on page 15 has also been copied below for convenience.

“Previous work examining the relationship between different measures of SES have found that household income, EA, occupational prestige and social deprivation all draw on similar underlying heritable traits. Despite this general genetic factor of SES, our study demonstrates that trait-specific loci are also evident, indicating that income and educational attainment capture heritable traits unique to each of them. Specifically, we estimate that 16% of the genetic variance of income is not shared with EA. The relevance of these income-specific genetic effects is underscored by several diverging relationships to health outcomes between EA and the genetic components of income not shared with EA (Non-EA income Factor). For example, the genetic correlation with schizophrenia differs between income and EA (income and schizophrenia $r_g = -0.04$, $SE = 0.02$, educational attainment and schizophrenia $r_g = 0.06$, $SE = 0.02$, **Supplementary Table 23**). This divergence is even stronger when the Non-EA Income Factor is considered (schizophrenia and NonEA Income $r_g = -0.23$, $SE = 0.04$). Furthermore, we found negative genetic correlations of the NonEA Income Factor with bipolar disorder, autism, and obsessive-compulsive disorder, while EA exhibits positive genetic correlations with these psychiatric outcomes. This may indicate that the educational system is more accommodating to individuals with these disorders than the labour market and/or that talents associated with these genetic risks (e.g., higher IQ with autism⁴⁸ or creativity with bipolar disorder and schizophrenia⁴⁹) are more advantageous in school than in the labour market.

More generally, the genetic components of the NonEA-Income Factor showed weaker associations with better physical health and health-related behaviour, such as drinking and smoking, compared to EA. One possible interpretation of this finding is that better health outcomes of higher socioeconomic status in wealthy countries could be more due to their association with education rather than with income or wealth, consistent with findings from quasi-experimental studies.”^{45–47}

Specific Comments:

1. The rationale for choosing different meta-analysis methods (MTAG vs. METAL) should be clearly articulated. The decision to use MTAG for sex-stratified analyses, rather than combining sexes from the outset, requires further justification. Clarifying these methodological choices will enhance the transparency and reproducibility of the study.

In our preregistered Analysis Plan, we requested that cohort analysts provide sex-stratified GWAS results for income. This approach allowed us to examine potential differences in the genetic architecture of income between men and women. We did not ask for pooled-sex analyses, as these could be obtained during the meta-analysis stage.

To combine the sex-specific analyses, we chose MTAG over METAL. MTAG can account for sample overlap across GWAS, which is important if the male and female samples within the same cohort share relatedness. In contrast, METAL does not account for relatedness across samples,

leading to anti-conservative standard errors when sample overlap exists. However, using MTAG to combine results at the cohort level had limitations: (i) it would require a separate MTAG analysis for each cohort, and (ii) MTAG relies on LD score regression to estimate sample overlap, which can produce noisy results when working with small GWAS datasets.

Given these challenges, we first combined the cohort-level results separately for each sex using METAL, as there was no reason to expect sample overlap across different cohorts. After that, we used MTAG to combine the sex-stratified meta-analysis results, accounting for potential overlap between male and female samples within the same datasets. We assumed perfect genetic correlation and equal heritability across sexes, making this step equivalent to a meta-analysis of pooled-sex GWAS results at the cohort level.

In the final stage, we applied MTAG to combine meta-analyses of different income measures, allowing for unequal heritability across these measures. This adjustment was necessary because different income metrics likely involved varying degrees of measurement error. MTAG's ability to account for sample overlap and differing measurement precision was essential at this step, particularly since some individuals appeared in multiple GWAS analyses with different income measures (e.g., the UKB sample had household income and occupational income measures, resulting in overlapping samples between both GWAS).

We have rewritten Section 2.5. of the Supplementary Information to clarify this.

2. The sentence "the estimated effects at the 5th and 95th percentiles were 0.18 and 0.60%, respectively" is ambiguous. It is unclear whether these percentages refer to the proportion of variance explained or another metric. Providing a more detailed explanation and context for these figures is needed.

We have clarified this sentence. It now reads as follows:

"The estimated effects at the 5th and 95th percentiles of the SNP effect size distribution were 0.18 and 0.60%, respectively."

3. When translating genetic effects into real dollar amounts (e.g. an annual income increase of \$169), provide the standard deviation for these translations.

We have rewritten the respective sentence as follows:

"A 0.3% increase would equal an additional annual income of \$169 (95% CI: \$102, \$339)."

4. Further discussing the potential future research directions, such as disentangling the interplay between genetic, behavioral, and environmental factors, will underscore the importance of this work.

We added the following paragraph to the discussion:

“Our results contribute to the understanding of genetic and environmental factors that influence income. Future research could focus on disentangling these relationships further by integrating genomic data with longitudinal assessments of environmental exposures and behavioral traits. Such approaches could help elucidate the pathways through which genetic predispositions interact with socioeconomic contexts, life experiences, and individual behaviors to shape income-related outcomes. This line of research may ultimately contribute to a deeper understanding of the mechanisms underlying social mobility and economic inequality.”